# LARGE LANGUAGE MODEL PROMPT DATASETS: AN IN-DEPTH ANALYSIS AND INSIGHTS

## ABSTRACT

A prompt is a natural language instruction that defines a specific task for a large language model (LLM) and serves as the primary interface for human-LLM interaction. With the growing deployment of LLMs, diverse prompt datasets are emerging from platforms such as GitHub and social media. These datasets span a wide array of applications and content types, facilitating both broader LLM utilization and improved prompt engineering. In this work, we–for the first time–have compiled an extensive list of prompt datasets sourced from various channels, representing a spectrum of downstream tasks, languages, engineering techniques, attributes, and modalities. We select key representative datasets for systematic analysis, revealing commonalities and differences in prompt construction across categories, distinguishing them from other text corpora like literature and web. We further propose a prompt optimization approach that leverages syntactic embeddings of part-of-speech and dependency structures. By identifying a centroid representation of prompts and guiding LLMs to rewrite prompts toward this centroid, our method improves the meaningfulness of model outputs. We have made our datasets and code available at `https://anonymous.4open.science/r/LLM-Prompt-Datasets-7416`.

## 1 INTRODUCTION

Recent large language model (LLM) advancements have spurred the proliferation of custom prompts optimized for specific tasks. This trend spans technology communities–from GitHub repositories (e.g., f/awesome-chatgpt-prompts (Akın)) and Reddit forums (e.g., ChatGPTPromptGenius (Cha)) to platforms like PromptBase and PromptGenius (Pro, a). AI researchers and domain experts also share prompts to promote transparency, reproducibility, and collaborative innovation (Conover et al., 2023; Chen et al., 2024). Collectively, these datasets enable detailed analysis of usage patterns and high-performing prompt designs.

Notably, prior research has largely neglected comprehensive examinations of available prompt datasets. To address this gap, we apply stringent criteria to select, refine, and evaluate datasets that enable analysis of diverse prompts across multiple sources, content types, and target applications. Our survey encompasses over 1.22 TB of data, comprising more than 673M prompt instances from 129 heterogeneous sources. Our first contribution is a hierarchical taxonomy of LLM prompt datasets that serves as a detailed reference for researchers and informs future studies.

Next, we perform multi-level linguistic analysis—lexical, syntactic, and semantic—across seven meticulously selected, large-scale, diverse, and representative prompt datasets. By integrating statistical and machine learning methods, our study reveals key insights into compositional patterns, domain-specific variations, and unique linguistic properties that distinguish these prompts from other text corpora, such as literature and web content.

Finally, we propose a prompt optimization method leveraging part-of-speech and dependency embeddings. By aligning target prompts with a centroid of high-performing syntactic patterns, our approach improves the meaningfulness and quality of LLM responses. This data-driven method provides a foundation for more effective prompt selection and refinement in LLMs.

## 2 RELATED WORK

**Datasets for LLMs.** Liu et al. (Liu et al., 2024) discuss broadly the topic of datasets for LLMs, but emphasize more on corpus datasets for training and fine-tuning LLMs rather than providing a detailed analysis of prompt datasets. A few works consider LLM prompt datasets but with a narrower objective compared to ours. For instance, (Zhang et al., 2023) and OpenCodeInstruct (Ahmad et al., 2025) focus only on datasets for instruction tuning–a technique for fine-tuning LLMs using carefully constructed instruction-response pairs. LLMSecEval (Tony et al., 2023) introduces a prompt dataset specifically designed for evaluating the safety of codes generated by LLMs, whereas Lu et al. (2024) survey datasets for LLMs' evaluation.

**Tools and frameworks for prompt engineering.** The prompt report (Schulhoff et al., 2024) offers a thorough survey of prompt engineering techniques, providing detailed scheme definitions and corresponding examples. Several works focus on developing tools that streamline prompt construction. Both PromptAid (Mishra et al., 2025) and PromptLandscape (Wang et al., 2024a) present visual support systems to simplify the creation and engineering of prompts. PEPR (Feffer et al., 2024) assesses various prompt combinations to determine the most optimal one for a given scenario. Saletta & Ferretti (2024) introduce a grammar-based evolutionary method to systematically optimize prompts for specific use cases. Promptaware (Chen et al., 2025) integrates software engineering principles into the prompt engineering process. There are also works proposing solutions for generating prompts for specific scenarios. PromptAgent (Wang et al., 2024b) introduces a model that automatically crafts and optimizes prompts with quality on par with those handcrafted by experts.

In contrast to previous studies, our work is the first to compile a comprehensive list of prompt datasets and we also extract valuable insights through their analysis.

## 3 PROMPT DATASETS DISCOVERY AND REFINEMENT

**Data discovery guideline.** We employ a systematic dataset discovery process across multiple sources to compile a diverse repository of prompt datasets. Our objective is to capture real-world, user-generated prompts, instruction-following interactions, and domain-specific scenarios. In particular, our primary objectives for datasets discovery are three-fold: **(1)** collecting datasets that are composed of prompts, i.e., natural language instructions that describe a certain task the LLM should perform and guide the LLM towards generating a desired output; **(2)** ensuring that the extracted data cover various domains, including day-to-day scenarios such as travel planning, professional scenarios such as academic writing, and specialized scenarios such as healthcare and finance; and **(3)** allowing different forms of prompts, e.g., single instruction, conversations, etc.

**Data discovery process.** We collect publicly available datasets from the following four types of sources. **First**, we consult *dataset collection platforms*, including Hugging Face Datasets (hug), Kaggle (kag), Google Dataset Search (goo), and Papers with Code (pap). Targeted searches using keywords, e.g., "prompt dataset", "instruction-following dataset", and "conversation dataset" yield 60 prompt datasets. **Second**, we review the latest *academic publications*, specifically papers on prompt engineering, natural language understanding, and dialogue systems, published at NeurIPS, ICLR, and ICML between 2023-2024 and identify 73 datasets shared across them. **Third**, we also examine *public repositories* by systematically surveying open-source GitHub projects using keywords, e.g., "prompt collection", "LLM prompts", and "instruction dataset". We identify 21 prompt repositories that typically contain curated prompt lists derived from user interactions or synthesized from public APIs. Some of these repositories are "awesome-lists", which are curated collections of high-quality prompts or links to prompt datasets. Notable examples include Awesome Instruction Datasets (Nie), Prompt Engineering Guide (Saravia, 2022), and LLMDataHub (Zhao). **Finally**, we extract 14 datasets from *popular websites dedicated to prompt-sharing*, including Prompt Genius (Pro, b) and BoredHumans (bor). These platforms feature user-written prompts for practical purposes.

**Data filtering.** We remove duplicate entries (e.g., CVQA (Romero et al., 2024) appears in both Hugging Face and NeurIPS 2024) and then filter the remaining candidates using four quality criteria for inclusion in this paper. **First**, *Dataset size.* We prioritize datasets containing at least 1K prompts to ensure robustness in diversity and statistical power. In contrast, due to their generally limited scope, user-shared datasets are filtered with a minimum threshold of 50 prompts. **Second**, *Data quality.* We evaluate the quality of prompts based on their cleanliness. Most datasets (e.g., OpenCodeReasoning

(Wasi Uddin Ahmad, 2025)) on data hosting platforms (e.g., Hugging Face and Kaggle) are well-formatted and clean. For the remaining data, we exclude samples with inconsistent formatting or unclear structure. For instance, the Prompt Engineering Guide–a resource that offers both curated prompt datasets and instructional examples–contains many illustrative prompts scattered throughout the material and are thus omitted from our datasets. **Third**, *Data relevance.* We assess whether the prompts are aligned with our data discovery guidelines, specifically emphasizing on those that represent common usage scenarios for broad audiences (e.g., Chinese-DeepSeek-R1-Distill-data-110k (Liu et al., 2025a)), and tasks from various domains (e.g., Medical Verifiable Problems (Chen et al., 2024), OpenMathReasoning (Moshkov et al., 2025)). Datasets that violate our discovery guidelines are omitted. For instance, the PersonaHub dataset (Per) is excluded because it does not meet **data discovery guideline (1)**, which mandates that prompts be linked to specific, well-defined tasks. Although PersonaHub demonstrates the potential of synthetic personas in generating diverse content (e.g., reasoning problems, dialogues, or non-player character behaviors), it predominantly comprises persona descriptions without clear task formulation. **Fourth**, *Accessibility.* Datasets must be publicly accessible or retrievable via automated crawling, and their licensing terms must permit research use. After filtering, we identify 129 distinct prompt datasets for taxonomic analysis (§4).

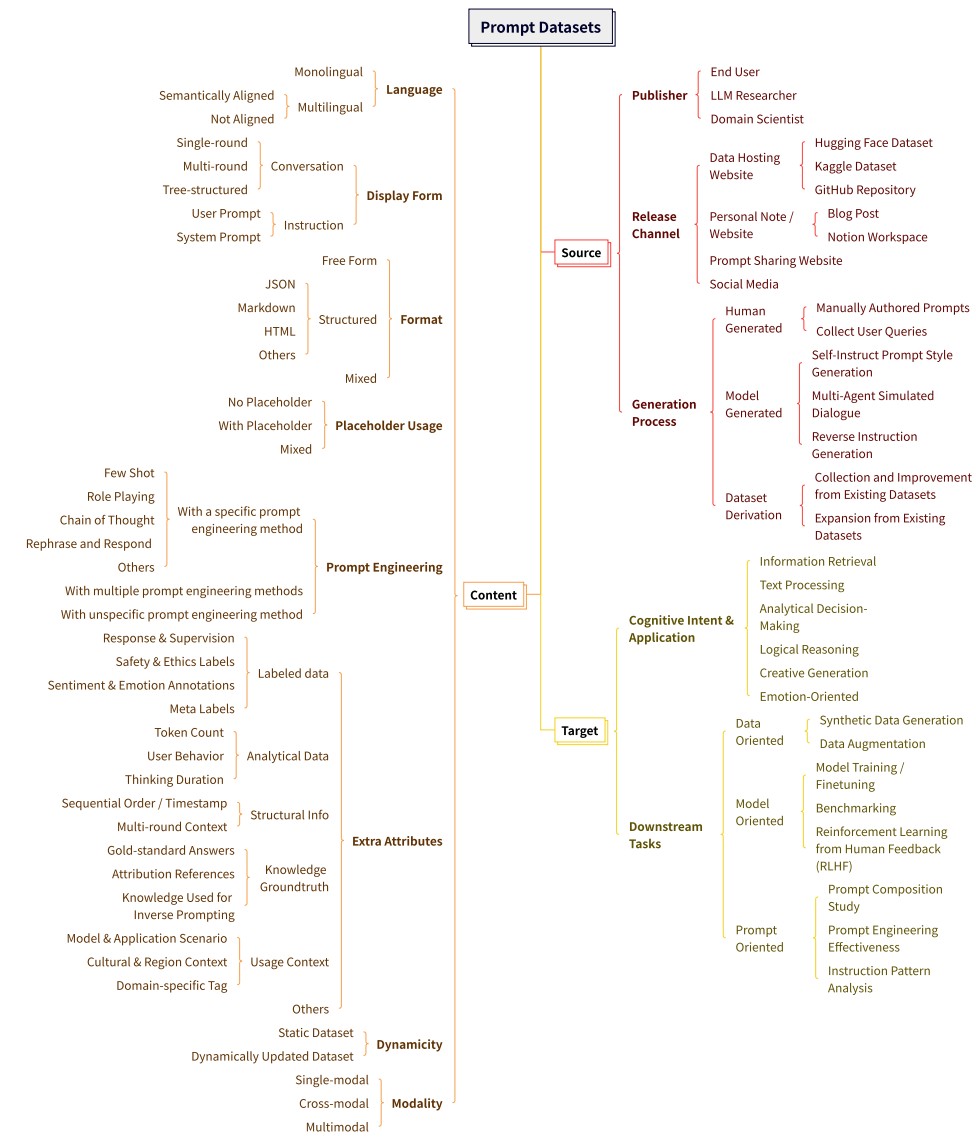

Figure 1: The hierarchical taxonomy of prompt datasets

## 4 DATASET TAXONOMY

We categorize our collected prompt datasets across multiple dimensions and hierarchies, creating a detailed taxonomy illustrated in Figure 1. We discuss certain key aspects in this taxonomy below.

**Source.** We classify prompt sources by publisher, release channel, and generation process.

**Publisher** denotes the source's identity and intent. We distinguish among *end users* who share prompts for practical tasks like writing/ coding (e.g., Prompt Genius), *LLM researchers* who publish prompts for fine-tuning and benchmarking (e.g., OpenMathReasoning by NVIDIA), and *domain scientists* who use LLMs in their specific fields (e.g., ChatGPT Data Science Prompts (Tang)).

**Release channel** refers to the platform where a dataset is published. Common platforms include *data hosting sites* such as GitHub, Hugging Face, and Kaggle, where structured prompt formats (e.g., CSV, JSON) dominate. *Personal sites* or *notes* (e.g., Notion workspaces) often host informal, user-oriented prompts. Dedicated *prompt sharing websites* vary from open-access (e.g., QuickRef.ME (Qui)) to commercial marketplaces (e.g., PromptBase). *Social media*, like Reddit's r/ChatGPTPromptGenius, also plays a key role in community-driven prompt exchange.

**Generation process** describes how the prompts are created. *Human-generated prompts* are either manually authored (e.g., databricks-dolly-15k (Conover et al., 2023)) or collected from user queries (e.g., ShareGPT (Li)), *Model-generated prompts* include those created via self-instruct techniques (Wang et al., 2023) (e.g., Self-Instruct), multi-agent simulations (e.g., AI Society (Li et al., 2023a)), or reverse instruction generation (e.g., LongForm (Köksal et al., 2023)). Finally, *derivative datasets* build on existing resources through task expansion or reformatted aggregation (e.g., Flan 2022 (Fla), xP3 (Muennighoff et al., 2022)).

**Content.** Prompt datasets are characterized by distinct linguistic and structural attributes. **Linguistically**, they may be *monolingual* or *multilingual*; in the latter case, datasets are deemed *semantically aligned* if each entry includes multilingual counterparts with identical semantics, thereby enhancing LLM performance (Li et al., 2023b). In terms of **display form**, prompts appear either as *conversation* (e.g., single-round, multi-round, or tree-structured) or *instruction* (e.g., user prompt, system prompt). The prompt **format**–ranging from *free-form* to *structured* (e.g., JSON, Markdown, HTML), or a combination thereof–substantially influences LLM response quality (Liu et al., 2025b). Finally, datasets differ in their use of **placeholders**, which allow for text substitution and enable diversified prompt transformations (Shin et al., 2020).

**Prompt engineering** methods are critical for enhancing prompt performance (Sahoo et al., 2025). Common techniques include *few-shot* (Brown et al., 2020), *role playing* (Zhang et al., 2018), *chain-of-thought* (CoT) (Wei et al., 2022), and *rephrase-and-respond* (Deng et al., 2023). Some datasets adopt a single method (e.g., awesome-chatgpt-prompts uses role playing), while others combine multiple techniques (e.g., PromptBench (Zhu et al., 2024) integrates six methods), or leave the strategy unspecified. Moreover, datasets may include **extra attributes**, including *labeled data* (e.g., response supervision, safety labels), *analytical data* (e.g., token count, user behavior), *structural information* (e.g., timestamp, multi-turn context), and *ground truth* (e.g., gold answers, attribution references)–as seen in datasets including hh-rlhf (Bai et al., 2022), UltraFeedback (Cui et al., 2023), and databricks-dolly-15k. *Additional tags* may indicate *usage context*, such as associated models, cultural regions, or domain specificity. Finally, datasets vary in **dynamicity** (i.e., static vs. dynamically updated) and **modality** (i.e., single-modal vs. cross-/multi-modal). For example, awesome-chatgpt-prompts is a dynamically updated prompts collection, while PLM-Video-Human (Cho et al., 2025) supports multi-modal learning for video understanding.

**Target.** Target defines the purpose and applications of prompt datasets. From **cognitive intents and applications** perspective, prompts may aim for *information retrieval* (e.g., databricks-dolly-15k includes information extraction category), *text processing* (e.g., StrategyQA (Str) requires implicit reasoning steps in the question), *analytical decision-making* (e.g., medical-o1-reasoning-SFT (Chen et al., 2024) for consultation decision), *logical reasoning* (e.g., DeepSeek-Prover-V1 (Xin et al., 2024)), *creative generation* (e.g., No Robots (Rajani et al., 2023) includes generation category), or *emotion-oriented* (e.g., empathetic-dialogues-facebook-ai (Emp)) tasks. In terms of **downstream tasks**, we identify three major categories: *data-oriented* (e.g., synthetic data generation, data augmentation), *model-oriented* (e.g., model training, finetuning, benchmarking, RLHF),

and *prompt-oriented* (e.g., prompt composition studies, prompt engineering effectiveness analysis, instruction pattern analysis).

Among these, instruction fine-tuning datasets represent a prominent and widely-used subset of prompt datasets (Liu et al., 2024). These datasets comprise instruction-response pairs, where the "instruction" serves as a prompt and the "response" represents the target model output. They are primarily employed for supervised fine-tuning to enhance model capability and controllability. As a result, models trained on these datasets exhibit superior alignment with human intent, improved instruction-following, and increased safety characteristics (Zhang et al., 2023). Furthermore, the instructions in these datasets often reflect real-world user queries, making them both practical for deployment and valuable for prompt-related research. Notable examples are Alpaca (Taori et al., 2023), OASST1 (Köpf et al., 2023), and FLAN 2022.

We summarize the key charecetristics and metadata attributes– including sources–of all our collected and filtered 129 prompt datasets in Appendix E.

## 5 PROMPT DATA ANALYSIS

We next conduct an in-depth analysis across three linguistic levels—lexical, syntactic, and semantic—of prompts derived from seven distinct sources. Our approach integrates statistical techniques with machine learning methods to identify compositional patterns and inter-source variations.

### 5.1 DATASET SELECTION

In order to ensure reliable analysis of prompt characteristics, we curate multiple prompt-centric datasets with the following selection principles. **(1)** *Language consistency.* Only English-language data have been included to ensure uniform linguistic features and avoid cross-linguistic biases. **(2)** *Exclusion of benchmark-style prompts.* Prompts designed for LLM performance evaluations (e.g., PHYBench (Qiu et al., 2025)) are excluded to focus on natural usage scenarios. **(3)** *Source and content diversity.* To achieve sufficient coverage and reduce sampling bias, we have selected datasets that differ in *publisher type* (i.e., end user vs. LLM researcher and domain scientist), *instruction generation method* (i.e., human vs. model generated), and *domain scope* (i.e., general vs. domain-specific tasks).

Following our selection principles, we curated seven representative datasets spanning different user types, instruction methods, and domains. For **end users**, general-domain prompts include single-turn prompts (BoredHumans) and multi-turn conversations (ShareGPT), while business-domain single-turn prompts are represented by 1100+ ChatGPT Prompts for Business. For **LLM researchers**, human-generated datasets include databricks-dolly-15k and OASST1, and model-generated prompts are captured by Self-Instruct. For **domain scientists**, we include model-generated medical prompts from medical-o1-reasoning-SFT. This collection ensures diversity in publisher type, prompt structure, and application domain.

- 1100+ ChatGPT Prompts for Business (**1.1k-business**). A curated dataset of 1 235 prompts oriented toward professional and business-related use cases, such as marketing, productivity, and decision-making. It represents structured, domain-specific prompting behavior (1.1).

- BoredHumans Prompts (**BoredHumans**). A smaller collection of 964 prompts compiled from publicly shared prompts on the boredhumans.com website. Some of the prompts on this site come from other community shared sources (e.g., awesome-chatgpt-prompts). It reflects community-created content and captures user creativity and experimentation (bor).

- databricks-dolly-15k (**dolly-15k**). This dataset includes 15 000 human-authored instruction–response pairs covering a range of everyday tasks. It is single-turn and domain-general, curated to support instruction-following models (Conover et al., 2023).

- medical-o1-reasoning-SFT (**medical-o1**). Synthetic data of 90 120 open-ended questions and GPT-4o generated CoTs and responses. Open-ended questions are reformatted by GPT-4o based on close-set medical examination questions. The dataset is used to fine-tune HuatuoGPT-o1 (Chen et al., 2024).

- OASST1. The Open Assistant dataset (OASST1) contains over 30 000 human-written messages arranged in dialogue trees. It emphasizes cooperative, open-domain assistant behavior and includes branching conversations rather than linear interactions (Köpf et al., 2023).

- Self-Instruct. A synthetic dataset with 82 646 prompts generated by large language models based on a small seed pool of human-written instructions. For every generation step, it samples 6 human-written tasks and 2 model-generated tasks in previous steps to promote diversity (Wang et al., 2023).

- ShareGPT. A large-scale collection of approximately 90 000 ChatGPT conversation logs shared by users. It represents multi-turn, organically generated interactions and captures diverse user intentions in real-world usage scenarios (Li).

We present the key characteristics of these seven datasets in Table 1.

Table 1: Key characteristics of the seven datasets selected for analysis, where size represents the number of prompts after preprocessing removes incorrectly extracted or malformed entries.

| Dataset | Size | Publisher Type | Generation Method | Display Form | Domain |
|---|---|---|---|---|---|
| **1.1k-business** | 1235 | End User | Unknown | User Prompt | Business |
| **BoredHumans** | 956 | End User | Dataset Derivation | User Prompt | General |
| **dolly-15k** | 14779 | LLM Researcher | Human Generated | Single-turn Conversation | General |
| **medical-o1** | 19679 | Domain Scientist | Model Generated | Single-turn Conversation | Medical |
| **OASST1** | 22079 | LLM Researcher | Human Generated | Tree-structured Conversation | General |
| **Self-Instruct** | 81673 | LLM Researcher | Model Generated | Single-turn Conversation | General |
| **ShareGPT** | 181570 | End User | Human Generated | Multi-turn Conversation | General |

## 5.2 TOKEN-LEVEL ANALYSIS

We perform token-level analysis using $n$-gram models to capture local textual patterns (Jurafsky & Martin, 2000; Cavnar & Trenkle, 1994; Manning & Schutze, 2001). Initially, all tokens are lemmatized to mitigate inflectional variability, after which we extract 3-gram, 4-gram, and 5-gram sequences to compute their frequency distributions. By analyzing high-frequency $n$-grams, we identify prevalent instruction templates, keyword combinations, and syntactic patterns, laying the groundwork for subsequent syntactic and semantic investigations.

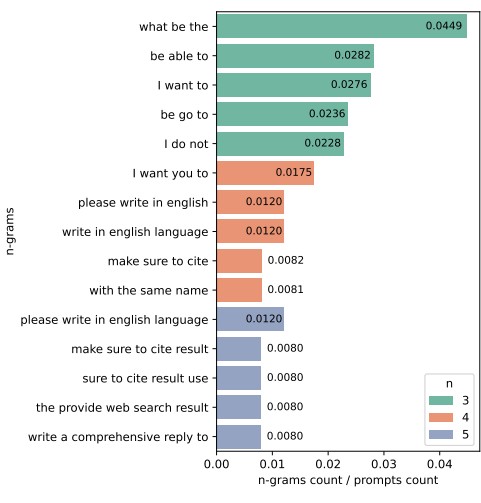
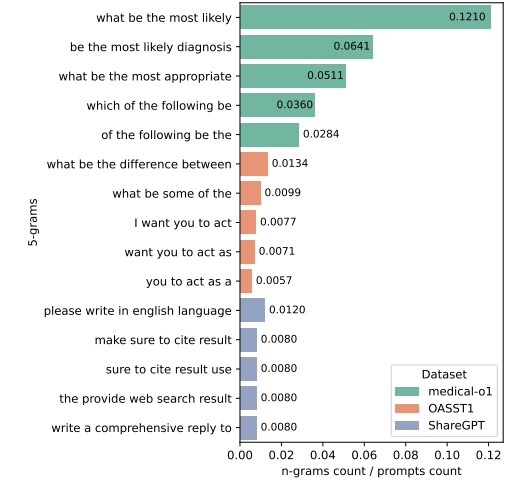

(a) Top-5 $n$-grams of ShareGPT ($n$=3, 4, 5)   (b) Top-5 5-grams of medical-o1, OASST1, ShareGPT

Figure 2: Comparison of 3/4/5-grams in the same dataset and 5-grams across multiple datasets. The ratio is defined as the count of the specific $n$-gram divided by the count of prompts in the dataset. More comprehensive comparison data and analysis can be found in Appendix F.1.

**Analysis of results.** The $n$-gram frequency distributions reveal several notable patterns that highlight the distinct functional and stylistic characteristics across datasets.

**(1)** High-frequency $n$-grams reveal domain and prompt-engineering differences, such as role-playing cues in OASST1 ("you to act as") versus medical reasoning in medical-o1 ("what be the," "the most likely diagnosis"). **(2)** While 3-grams capture general-purpose queries or commands (e.g., "what

be the," "I want to"), longer $n$-grams (4–5) reflect task-specific patterns, as in ShareGPT where frequent 5-grams ("please write in English language," "write a comprehensive reply to") highlight its instruction-following orientation. **(3)** Compared to Google Books 5-grams (e.g., "at the end of the," "in whole or in part") that serve narrative or descriptive purposes, prompt datasets exhibit inquiry- or command-focused $n$-grams, underscoring a clear divergence in linguistic patterns across corpora.

## 5.3 SYNTACTIC-LEVEL ANALYSIS

To gain deeper insights into the linguistic structure of prompts, we perform syntactic analysis from three perspectives: dependency parsing (Nivre, 2003), part-of-speech (POS) tagging (Brill, 1992), and term frequency-inverse document frequency (TF-IDF) scoring (Salton & Buckley, 1988). These features are both descriptive and can be aggregated into vector representations for tasks like prompt classification.

For comparative analysis with non-prompt text datasets, we have used Universal Dependencies corpora for English: EWT (Silveira et al., 2014) and ParTUT (Sanguinetti & Bosco, 2014), where EWT contains informal contents–blog, social, reviews, email, and web, and ParTUT contains more formal contents–legal, news, and wiki.

Table 2: Top-8 dependency types, with the values indicating their proportions in the dataset. The dependency types represent syntactic relationships between words in a sentence: **punct**–punctuation marks; **prep**–prepositions; **det**–determiners (e.g., "the", "a"); **pobj**–prepositional objects; **dobj**–direct objects; **nsubj**–nominal subjects; and **ROOT**–the sentence's main verb or predicate. Note that spaCy's (`en_core_web_sm`) dependency labels do not entirely conform to the Universal Dependencies standard; non-conforming labels are represented with a dash ("-") in cross-corpus comparisons. Full data in Table 5.

| Dependency Type | EWT | ParTUT | 1.1k-business | BoredHumans | dolly-15k | medical-o1 | OASST1 | Self-Instruct | ShareGPT |
|---|---|---|---|---|---|---|---|---|---|
| punct | 0.12 | 0.12 | 0.1227 | **0.1985** | 0.1445 | 0.1216 | 0.1273 | 0.1863 | 0.1540 |
| prep | - | - | 0.0759 | 0.0672 | 0.0866 | **0.1013** | 0.0816 | 0.0676 | 0.0764 |
| det | 0.08 | 0.09 | 0.0518 | 0.0692 | **0.0961** | 0.0906 | 0.0841 | 0.0838 | 0.0693 |
| pobj | - | - | 0.0718 | 0.0620 | 0.0817 | **0.0979** | 0.0760 | 0.0645 | 0.0711 |
| nsubj | 0.08 | 0.06 | 0.0596 | 0.0545 | 0.0650 | 0.0469 | **0.0739** | 0.0596 | 0.0562 |
| ROOT | 0.07 | 0.04 | 0.0528 | 0.0462 | 0.0768 | 0.0444 | 0.0604 | **0.0792** | 0.0437 |
| amod | 0.05 | 0.06 | 0.0573 | 0.0527 | 0.0469 | **0.1072** | 0.0523 | 0.0384 | 0.0480 |
| dobj | - | - | **0.0904** | 0.0665 | 0.0447 | 0.0315 | 0.0594 | 0.0570 | 0.0519 |

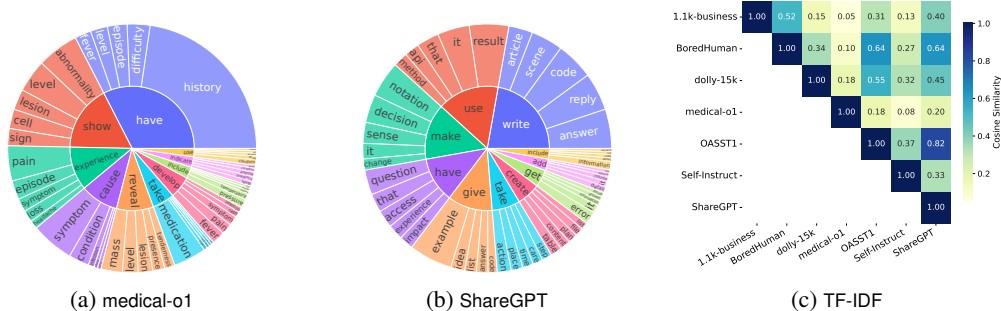

| (a) medical-o1 | (b) ShareGPT | (c) TF-IDF |

Figure 3: (a-b): The top-10 most common verbs and their top-5 direct noun objects in two prompt datasets. Data for other 5 datasets are shown in Figure 8. (c): Cosine similarity between dataset-level TF-IDF vectors.

### 5.3.1 DEPENDENCY PARSING

We apply the spaCy `en_core_web_sm` parser (Honnibal & Montani, 2017) to extract syntactic dependencies and determine the frequency of key grammatical relations in each dataset. For the EWT and ParTUT corpora, we rely on officially published dependency type annotations. This analysis reveals systematic variations in linguistic style across prompt sources. Additionally, we track verb–object (dobj) pairs to capture the task-oriented diversity of the prompts (see Figure 3).

**Analysis of Results.** Table 2 shows the distribution of eight common dependency types across seven prompt datasets and two reference corpora (EWT and ParTUT), revealing three key findings.

**(1)** The medical-o1 dataset is characterized by its high use of adjectival modifiers (amod, 0.11) and low direct object frequency (dobj, 0.03), reflecting a preference for precise, state-oriented descriptions over action-driven narratives, often framed through linking verbs—typical of medical

Table 3: The top-7 Parts-of-Speech, with each value indicating its proportion in a dataset. Full data in Table 6.

| POS | EWT | ParTUT | 1.1k-business | BoredHumans | dolly-15k | medical-o1 | OASST1 | Self-Instruct | ShareGPT |
|---|---|---|---|---|---|---|---|---|---|
| NOUN | 0.17 | 0.21 | **0.2637** | 0.2103 | 0.1899 | 0.2590 | 0.1946 | 0.2027 | 0.1944 |
| PUNCT | 0.12 | 0.12 | 0.1094 | **0.1942** | 0.1435 | 0.1158 | 0.1231 | 0.1839 | 0.1450 |
| VERB | 0.11 | 0.10 | **0.1302** | 0.1094 | 0.0871 | 0.0775 | 0.1069 | 0.0999 | 0.0979 |
| ADP | 0.09 | 0.12 | 0.0758 | 0.0678 | 0.0858 | **0.0998** | 0.0851 | 0.0701 | 0.0789 |
| DET | 0.08 | **0.11** | 0.0506 | 0.0693 | 0.0949 | 0.0893 | 0.0839 | 0.0844 | 0.0696 |
| PRON | 0.09 | 0.04 | **0.0912** | 0.0708 | 0.0695 | 0.0369 | 0.0870 | 0.0701 | 0.0583 |
| ADJ | 0.07 | 0.08 | 0.0588 | 0.0543 | 0.0538 | **0.1104** | 0.0632 | 0.0498 | 0.0563 |

contexts detailing conditions, symptoms, and diagnoses. **(2)** In contrast, the 1.1k-business dataset favors concise, goal-driven imperatives with bare noun phrases as direct objects (dobj, 0.09) and minimal use of determiners (det, 0.05), aligning with its project-planning focus. **(3)** Verb–noun dependency analysis further distinguishes domains: medical instructions cluster around technical, domain-specific pairs like "have history" and "experience pain," while datasets such as ShareGPT use broader, generic pairs like "write answer" and "use code". These syntactic patterns highlight each corpus' thematic priorities and inform strategies for domain-aware model training.

These findings highlight the stylistic diversity among prompt datasets, where domain and intent directly influence grammatical structure. Medical prompts stress detailed specificity and descriptive richness, while business prompts favor concise, directive clarity, illustrating the functional interplay between form and purpose.

### 5.3.2 PART-OF-SPEECH TAGGING

We annotate the datasets with POS tags and calculate the distribution of nouns, verbs, adjectives, and adverbs. Table 3 summarizes the functional composition of prompts, contrasting content and function words. For example, a high verb frequency indicates action-oriented prompts, while a predominance of nouns suggests more objective narratives. These distributional differences reveal stylistic and structural variations across sources.

**Analysis of results. (1)** Domain-specific datasets such as 1.1k-business and medical-o1 exhibit a noun proportion of $\approx 0.26$, surpassing that found in formal corpora like ParTUT. This reflects a concept-driven focus on domain entities and technical terms. **(2)** Additionally, medical-o1 also registers an unusually high adjective ratio (0.11), indicating a repeated emphasis on specifying medical attributes and conditions, consistent with the descriptive nature of clinical reasoning tasks.

### 5.3.3 TF-IDF ANALYSIS

We analyze lexical patterns across prompt datasets using TF-IDF. Each dataset's prompts are concatenated into a single document (yielding seven corpus-level documents), and a TF-IDF vectorizer (with a 5000-word limit and English stopwords removed) computes sparse term importance representations. We then assess **inter-dataset lexical similarity** via pairwise cosine similarity (Figure 3c) and extract the top three highest-weight tokens per dataset for **intra-dataset characterization** (Table 4).

Table 4: Top-3 tokens with the highest TF-IDF weights per dataset

| Dataset | Top-3 tokens |
|---|---|
| 1.1k-business | content (0.308), email (0.284), marketing (0.245) |
| BoredHumans | act (0.269), want (0.261), write (0.217) |
| dolly-15k | list (0.338), given (0.246), following (0.241) |
| medical-o1 | old (0.427), year (0.367), patient (0.256) |
| OASST1 | write (0.307), like (0.216), does (0.207) |
| Self-Instruct | output (0.766), input (0.292), task (0.243) |
| ShareGPT | write (0.190), use (0.180), data (0.157) |

**Analysis of results. (1) Intra-dataset analysis** delineates each dataset's lexical focus and stylistic characteristics. For instance, 1.1k-business emphasizes business-specific terms like "content" and "email", while BoredHumans features imperatives such as "act", indicative of role-playing instructions. Similarly, Self-Instruct shows a dominant TF-IDF score for "output" (0.772), highlighting a structural prompt style based on explicit instruction–response formats. **(2) Inter-dataset comparison.** TF-IDF vectors show varying overlaps across datasets. The highest cosine similarity between OASST1 and ShareGPT suggests a similar vocabulary—likely due to shared human-generation processes. In contrast, Self-Instruct is lexically distant from the others, especially 1.1k-business and medical-o1, reflecting stylistic and domain-specific differences.

### 5.4 SEMANTIC-LEVEL ANALYSIS

We analyze prompt semantics by encoding each prompt into a 384-dimensional dense vector using Sentence-BERT's pretrained model `all-MiniLM-L6-v2` (Reimers & Gurevych, 2019). Each

prompt is encoded into a 384-dimensional dense vector that captures its semantic content. These embeddings serve as the foundation for classification, clustering, and visualization analysis. We perform Principal Component Analysis (PCA) to reduce sentence embeddings to two dimensions. For fair comparison, we uniformly at random sample 500 prompts per dataset and visualize their distribution (Figure 4).

**Analysis of results. (1) Wide coverage in Self-Instruct:** The Self-Instruct dataset exhibits the most dispersed and evenly distributed semantic space, suggesting a broad topical coverage. This aligns with the self-instruction paradigm's goal of generating diverse instruction types. **(2) Semantic cohesion in specific domains:** Prompts from medical-o1 and 1.1k-business form more

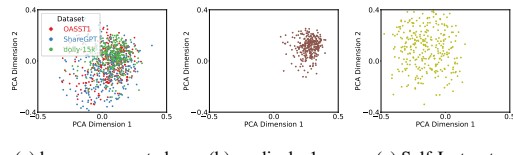

(a) human-generated    (b) medical-o1    (c) Self-Instruct

Figure 4: Semantic prompt embeddings distribution.

concentrated clusters, indicating domain-specific semantic cohesion. **(3) Overlap among human-generated sets:** The embeddings of dolly-15k, OASST1, and ShareGPT overlap substantially across both PCA dimensions. This suggests that these datasets share stylistic and semantic characteristics, possibly due to their common reliance on human-LLM interactions for data generation.

## 6 APPLICATION

Building on the above analysis, we propose a new prompt engineering method that leverages structural linguistic features. Specifically, we take the average of the high-dimensional embeddings of POS tags and dependency relations from the analyzed dataset to define a centroid representation. This centroid captures the "central" syntactic patterns that are associated with higher-performing prompts.

For each target prompt, we first analyze its POS and dependency embeddings to identify deviations from the centroid. Based on this analysis, a modification plan is generated, specifying how the prompt's syntactic structure should be adjusted. The LLM is then guided to rewrite the prompt according to this plan, producing an optimized prompt whose embeddings are closer to the centroid. This process allows peripheral prompts that initially deviate from effective syntactic patterns to be systematically aligned with the central region of the embedding space.

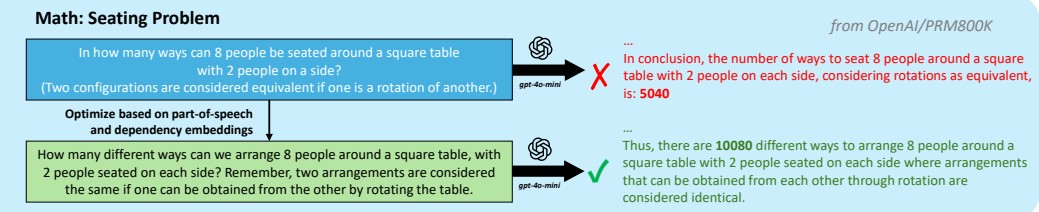

Figure 5: A case study of prompt optimization

To illustrate the practical impact of this approach, we present one representative case study in Figure 5, where our prompt optimization successfully corrected the model's initial incorrect responses, while another case study and the complete text are provided in Appendix F.4.

By aligning prompts with this centroid, our method aims to improve the likelihood that the LLM generates correct or more meaningful responses.

## 7 CONCLUSIONS

We addressed the underexplored challenge of collecting and categorizing LLM prompt datasets into a structured taxonomy. Our lexical, syntactic, and semantic explorations uncover key linguistic patterns, inter-dataset similarities, differences, and distinctions from other corpora such as literature and web content. Our novel application enhances domain-specific prompt filtering pipelines by automatically flagging irrelevant or malformed prompts in an unsupervised, data-driven manner before inference. Future work should leverage these extensive datasets to advance LLM architectures, prompt engineering, and human-AI interactions—including adaptive quality assessments and pricing models in AI prompt marketplaces.

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

## A    LIMITATIONS AND DISUCUSSION

While our analysis offers valuable insights, several limitations remain. Below, we outline these limitations and discuss their potential impact on our findings.

**Limited Datasets for Analysis**    To facilitate a focused analysis, we selected seven datasets representing various publisher types, generation methods, display formats, and domain categories. However, the number of datasets in each category is relatively limited, which may affect the representativeness of the results. Additionally, the scope of categories included in our analysis could be further expanded in future work to improve the comprehensiveness of the evaluation.

**Lack of Evaluation of Prompt Effects**    Due to the diversity of tasks among our selected datasets, this study does not include an evaluation of prompt effects. Our research primarily focuses on Natural Language Processing and Machine Learning methods, and thus does not leverage Large Language Models to analyze the impact of prompts. We recognize this as a limitation and suggest that future research incorporate prompt-based approaches for a more thorough assessment.

The amount of prompt data is growing rapidly. Based on our proposed taxonomy, we encourage further studies to continuously explore and update analyses in accordance with emerging trends. Additionally, we hope that future research will conduct more in-depth experiments on prompt datasets, which could be vital for advancing prompt design.

## B    LLM USAGE

LLMs are only involved in this work for grammatical checking and smart auto-completion during code implementation.

## C    ETHICS STATEMENT

Our work conforms to the ICLR Code of Ethics by responsibly compiling and analyzing existing prompt datasets rather than collecting new sensitive data. We ensure proper documentation of all datasets analyzed, respect original licenses and sources, and have made our code and datasets publicly available for transparency and reproducibility. The research poses minimal risk for misuse as it focuses on analytical insights rather than creating potentially harmful technologies, and we have documented our methodology thoroughly to enable external scrutiny.

## D    REPRODUCIBILITY STATEMENT

We are committed to ensuring the reproducibility of our results. All code used in this research is publicly available through links in our abstract. The repository includes detailed instructions for dataset preprocessing, and running experiments. We also specify the exact versions of dependencies and libraries used in our experiments. All datasets employed in this study are either publicly accessible or their sources are clearly documented. Random seeds are set for all experiments where applicable to minimize variability. Together, these resources enable researchers to reproduce our analyses and results with minimal effort.

## E    SUMMARY OF PROMPT DATASETS FOR TAXONOMIC ANALYSIS

We briefly discuss all 129 prompt datasets collected for taxonomic analysis (§3 and §4).

Note that the labeled license refers to the licensing information assigned to the dataset based on the publishers' declared rights. However, certain sub-datasets may remain subject to their original licensing conditions, which could differ from the labeled license.

1. 1100+ ChatGPT Prompts for Business
   - **Publisher**: Chris Porter

- **Size**: 1235 instances
- **License**: -
- **Link**: https://chatgpt-business-prompts.notion.site/1100-ChatGPT-Prompts-for-Business-eea03b0bc9b84ae7a5bdbd76a67460f3
- **Description**: "1100+ ChatGPT Prompts for Business" is a Notion-based dataset containing 1,235 curated prompts tailored for diverse business scenarios. It spans key domains such as buyer persona development, content strategy, digital marketing, narrative marketing, email campaigns, market research, product innovation, and finance. The collection includes specialized roles like Simulation Specialist, offering practical guidance for professionals, marketers, and entrepreneurs aiming to optimize operations, boost engagement, and enhance strategic decision-making.

2. 2.5k-chatgpt-promp-templates

- **Publisher**: TheVeller
- **Size**: 1088 instances
- **License**: -
- **Link**: https://ignacio-velasquez.notion.site/2-500-ChatGPT-Prompt-Templates-d9541e901b2b4e8f800e819bdc0256da
- **Description**: This dataset comprises over 1,000 curated ChatGPT prompt templates in Notion Workspace format, spanning diverse domains such as AI, marketing, education, healthcare, and code generation. Each entry typically includes a prompt, an automatic prompt (system prompt like), and a concise description.

3. A Collection of AI's Prompts for optimal context

- **Publisher**: Marc-Aurele Besner
- **Size**: 70 instances
- **License**: MIT
- **Link**: https://github.com/marc-aurele-besner/ChatGPT-PromptsList
- **Description**: This repository offers a well-curated collection of conversation prompts tailored for OpenAI's GPT-3 model.

4. Academic Reasoning and Intuition Chains Dataset

- **Publisher**: Marco De Santis
- **Size**: 2024 instances
- **License**: Apache-2.0
- **Link**: https://huggingface.co/datasets/marcodsn/academic-chains
- **Description**: The Academic Reasoning and Intuition Chains dataset comprises 1,975 examples of chain-of-thought reasoning distilled from open-access arXiv papers across eight scientific domains, including Biology, Economics, Physics, Mathematics, Computer Science, Finance, Statistics, and Electrical Engineering. Each entry contains comprehensive metadata (arxiv_id, DOI, authors, dates, and categories), interactive model-generated conversations with explicit <think> tags, extensive chain length statistics, and multi-model verifier results with suitability scores. Licensed under Apache-2.0, this resource enables training and evaluation of budgeted chain-of-thought reasoning models with rigorous quality control.

5. AI Short

- **Publisher**: rockbenben
- **Size**: 5867 instances
- **License**: -
- **Link**: https://www.aishort.top/
- **Description**: AI Short is a public prompt-sharing platform with 5,867 categorized prompts. Each prompt is available in multiple languages, enabling cross-linguistic studies of prompt effectiveness and translation consistency.

6. AI-Generated Prompts Dataset

- **Publisher**: Anthony Therrien

- **Size**: 173574 instances
- **License**: CC-BY-SA-4.0
- **Link**: `https://www.kaggle.com/datasets/anthonytherrien/ai-generated-prompts-dataset`
- **Description**: This dataset features thousands of prompts generated by the teknium/OpenHermes-2p5-Mistral-7B model, each designed to elicit diverse and contextually rich responses. Stored as JSON objects, it enables research in synthetic prompt generation, model creativity evaluation, and downstream fine-tuning.

7. AIPRM

- **Publisher**: AIPRM
- **Size**: 5325 instances
- **License**: -
- **Link**: `https://www.aiprm.com/`
- **Description**: AIPRM is a community-curated prompt library and management platform featuring 5,325 publicly accessible prompts categorized by topic and activity. Its user-driven structure offers valuable insights into real-world prompt usage, preferences, and task design patterns.

8. Alpaca_data

- **Publisher**: Stanford Alpaca
- **Size**: 52K instances
- **License**: Apache-2.0
- **Link**: `https://github.com/tatsu-lab/stanford_alpaca/tree/main`
- **Description**: The Stanford Alpaca dataset comprises 52K high-quality, instruction-following examples generated via a modified Self-Instruct pipeline using text-davinci-003. Designed for fine-tuning LLaMA models, it enables research in alignment, instruction tuning, and synthetic data generation.

9. Alpaca_GPT4_data_zh

- **Publisher**: Microsoft Research
- **Size**: 52K instances
- **License**: Apache-2.0
- **Link**: `https://huggingface.co/datasets/llm-wizard/alpaca-gpt4-data-zh`
- **Description**: Alpaca_GPT4_data_zh is a Chinese instruction-tuning dataset curated by the Instruction Tuning with GPT-4 project. It comprises 48,818 examples, each featuring an instruction, optional input context, and a GPT-4-generated response, facilitating text-generation and fine-tuning tasks. The dataset occupies 32 MB and is available under a CC-BY-4.0 license for non-commercial research.

10. AM-DeepSeek-Distilled-40M

- **Publisher**: a-m-team
- **Size**: 40M instances
- **License**: CC-BY-NC-4.0
- **Link**: `https://huggingface.co/datasets/a-m-team/AM-DeepSeek-Distilled-40M`
- **Description**: AM-DeepSeek-Distilled-40M is a multilingual (zh/en) reasoning dataset comprising 3.34 million prompts paired with 40 million model-generated responses across code, math, science, instruction-following and general reasoning. Each query includes four samples from three models (1.5B, 7B, and R1), with pass rates computed per model to assign unbiased difficulty scores. Released under CC-BY-NC 4.0, its unified JSONL format supports supervised fine-tuning, preference learning and reinforcement learning applications, enabling selection of subsets by category or difficulty level. It fosters robust LLM development research.

11. AM-DeepSeek-R1-Distilled-1.4M

- **Publisher**: a-m-team
- **Size**: 1.4M instances
- **License**: CC-BY-NC-4.0
- **Link**: `https://huggingface.co/datasets/a-m-team/AM-DeepSeek-R1-Distilled-1.4M`
- **Description**: AM-DeepSeek-R1-Distilled-1.4M is a bilingual (Chinese and English) reasoning dataset of 1.4 million challenging problem-solution pairs. Collected from diverse open-source sources, it features semantically deduplicated instructions spanning text, code, and math domains. It provides high-quality, comprehensive, and diverse reasoning challenges. Solutions are distilled mainly from DeepSeek-R1-671B and rigorously validated via test-case execution, answer checking, and reward-model scoring. Structured as user-assistant exchanges with reasoning traces and metadata, this cc-by-nc-4.0 dataset also offers 0.5M, 0.9M, and 1K-sample zstd-compressed configs to support scalable LLM research.

12. AM-Math-Difficulty-RL
    - **Publisher**: a-m-team
    - **Size**: 234729 instances
    - **License**: CC-BY-NC-4.0
    - **Link**: `https://huggingface.co/datasets/a-m-team/AM-Math-Difficulty-RL`
    - **Description**: AM-Math-Difficulty-RL is an English math dataset comprising three difficulty tiers designed for RL of LLMs. It contains 100k+ problems from repositories and categorized by pass rates of Qwen models. Tier 1 includes tasks with partial success by Qwen-1.5B; Tier 2 covers problems where smaller models fail but larger ones succeed; Tier 3 features examples that even Qwen-32B struggles with. Problems span algebra, calculus, and combinatorics. Licensed under CC-BY-NC-4.0, it supports text-generation tasks and research on difficulty-aware staged RL strategies.

13. APIGen-MT-5k
    - **Publisher**: Salesforce AI Research
    - **Size**: 5K instances
    - **License**: CC-BY-NC-4.0
    - **Link**: `https://huggingface.co/datasets/Salesforce/APIGen-MT-5k`
    - **Description**: The APIGen-MT-5k dataset comprises 5000 realistic, high-quality, multi-turn function-calling dialogues generated by APIGen-MT, a scalable automated agentic pipeline simulating agent-human interactions. Covering retail and airline domains, each trajectory is verified through format checks, function executions, and semantic validations, achieving a 99% success rate in human evaluation. Provided in ShareGPT-style JSON and licensed under CC-BY-NC-4.0, it supports question-answering, text generation, and reinforcement learning benchmarks.

14. awesome-chatgpt-prompts
    - **Publisher**: Fatih Kadir Akın
    - **Size**: 211 instances
    - **License**: CC0-1.0
    - **Link**: `https://github.com/f/awesome-chatgpt-prompts`
    - **Description**: The Awesome ChatGPT Prompts dataset is a collaboratively curated collection of diverse prompts optimized for interactive AI models, including ChatGPT, Claude, and LLaMA. Featuring both human- and LLM-generated entries with clear attribution, it supports research in prompt engineering, prompt effectiveness, and cross-model generalization.

15. Aya Collection
    - **Publisher**: Cohere For AI Community et al.
    - **Size**: 513M instances
    - **License**: Apache-2.0
    - **Link**: `https://huggingface.co/datasets/CohereLabs/aya_collection`

- **Description**: Aya Collection is a massive multilingual instruction tuning dataset comprising over 513 million prompt-completion pairs across 115 languages. It integrates three sources: human-crafted instruction templates created by fluent speakers for diverse tasks, machine translations of 19 top-tier datasets into 101 languages via NLLB, and the human-annotated Aya Dataset subset of 204K examples. Split by dataset, each record includes id, inputs, targets, language, script, and task type. Licensed under Apache-2.0, it supports academic and commercial classification, summarization, translation, and QA research.

16. Aya Dataset

    - **Publisher**: Cohere For AI Community et al.
    - **Size**: 204K instances
    - **License**: Apache-2.0
    - **Link**: `https://huggingface.co/datasets/CohereLabs/aya_dataset`
    - **Description**: The Aya Dataset is a multilingual, human-annotated instruction fine-tuning resource encompassing 204K prompt-completion pairs across 65 languages and dialects. It includes original annotations, re-annotations, and detailed annotator demographics such as age, gender, and regional background. Collected via the open-science Aya Annotation Platform, it supports diverse linguistic representation from high- to low-resource languages. Released under Apache 2.0, Aya is designed to train, fine-tune, and evaluate large language models on cross-cultural instruction following. It offers train (202K examples) and test splits with tasks.

17. BABILong

    - **Publisher**: AIRI et al.
    - **Size**: 25K instances
    - **License**: Apache 2.0
    - **Link**: `https://huggingface.co/datasets/RMT-team/babilong`
    - **Description**: BABILong is a generative benchmark designed to evaluate large language models' ability to perform reasoning over extremely long contexts. It embeds the ten bAbI tasks within irrelevant PG19 background text, creating "needle-in-a-haystack" scenarios across sequence lengths ranging from 0k to 1M tokens. Each task probes basic reasoning skills—such as supporting-fact retrieval, negation, and counting—amidst distractors. BABILong thus challenges models to identify pertinent facts and answer questions accurately.

18. Bactrain-X

    - **Publisher**: MBZUAI
    - **Size**: 3484884 instances
    - **License**: CC-BY-NC-4.0
    - **Link**: `https://huggingface.co/datasets/MBZUAI/Bactrian-X`
    - **Description**: Bactrian-X is a multilingual instruction-following dataset containing 3.4 million instruction-input-response triplets across 52 languages. It builds upon 67K unique English prompts drawn from Alpaca and Dolly, automatically translated via Google Translate into 51 languages. For each translated prompt (and optional input), GPT-3.5-Turbo generates a corresponding response, yielding 3.4 million examples. Each record includes an id, instruction, optional input, and model-generated output. Released under CC-BY-NC 4.0, Bactrian-X supports text-generation research, fine-tuning, and evaluation in low-resource and high-resource language settings, covering diverse tasks and domains.

19. Baize

    - **Publisher**: University of California et al.
    - **Size**: 210311 instances
    - **License**: GPL-3.0
    - **Link**: `https://huggingface.co/datasets/linkanjarad/baize-chat-data`
    - **Description**: Baize Chat Data is an instruction-finetuning corpus combining four sources: Alpaca, Medical, Quora, and StackOverflow. It contains about 210,000 conversational examples, each formatted with [|Human|] prompts and [|AI|] responses. Designed to enhance the

Baize family of language models, this unified dataset supports interactive text generation and dialogue training. Sourced from the Baize GitHub repository, it provides diverse conversational scenarios ranging from general queries to specialized medical and technical discussions. It is optimized for instruction-following tasks. It enables realistic user interactions.

20. BELLE_Generated_Chat

- **Publisher**: BELLE
- **Size**: 396004 instances
- **License**: GPL-3.0
- **Link**: https://huggingface.co/datasets/BelleGroup/generated_chat_0.4M
- **Description**: BELLE_Generated_Chat contains approx. 400k personalized Chinese character dialogues generated by the BELLE project. Each record includes an instruction, an (empty) input, and a generated output. Created by ChatGPT and not strictly verified, the dataset may contain factual inaccuracies. Licensed under GPL-3.0 for research use only. With around 0.4 million entries, it supports text-to-text generation and conversational modeling.

21. BELLE_Multiturn_Chat

- **Publisher**: BELLE
- **Size**: 831036 instances
- **License**: GPL-3.0
- **Link**: https://huggingface.co/datasets/BelleGroup/multiturn_chat_0.8M
- **Description**: BELLE_Multiturn_Chat is a Chinese multi-turn conversational dataset comprising approximately 0.8 million human-assistant dialogues generated by the BELLE project using ChatGPT. Each record pairs an instruction containing prior context labeled with "Human:" and "Assistant:" with the assistant's subsequent reply. Intended for text-to-text generation tasks, the GPL-3.0-licensed collection covers only Chinese interactions. As this data is automatically generated and unverified, factual errors and inconsistencies may arise. It is provided strictly for non-commercial research under the project's usage restrictions; developers should validate outputs and adhere to licensing terms.

22. BELLE_train_3.5M_CN

- **Publisher**: BELLE
- **Size**: 3606402 instances
- **License**: GPL-3.0
- **Link**: https://huggingface.co/datasets/BelleGroup/train_3.5M_CN
- **Description**: The BELLE_train_3.5M_CN dataset comprises approximately 3.5 million monolingual Chinese instruction-response pairs generated by the BELLE project, formatted as multi-turn and single-turn dialogues with unique IDs. It includes human-assistant exchanges across 13 instruction categories. Licensed under GPL-3.0, it supports text-to-text generation research exclusively; commercial or harmful use is prohibited. The JSON records each conversation's ID and bilingual content.

23. best-chinese-prompt

- **Publisher**: K-Render
- **Size**: 141 instances
- **License**: -
- **Link**: https://github.com/K-Render/best-chinese-prompt
- **Description**: The Best Chinese Prompt dataset is a comprehensive, well-structured collection of Chinese-language prompts spanning diverse categories such as casual chat, knowledge Q&A, creative planning, copywriting, and code generation. It provides real multi-model response comparisons (e.g., GPT-4, ChatGPT, NewBing, Wenxin) and continuous updates via collaborative platforms.

24. BigDocs-Bench

- **Publisher**: ServiceNow Research et al.
- **Size**: 415740 instances

- **License**: CC-BY-4.0
- **Link**: `https://huggingface.co/datasets/ServiceNow/BigDocs-Bench`
- **Description**: BigDocs-Bench is a CC-BY-4.0 benchmark suite for training and evaluating multimodal models on document and code tasks. It comprises seven configurations: GUI-VQA, GUI2BBox, GUI2Summary, GUI2UserIntent, Image2Flow (GraphViz/JSON), and Table2LaTex, each containing thousands of samples across train, validation, and test splits. Spanning over 7.6 TB with 200K+ annotated examples, it includes screenshots or generated images paired with queries, annotations, metadata, and optional filter flags. Auxiliary fields trace provenance and dependencies on arXiv, SeeClick, AFTdb, InternVL-8B, LLaMA 3.1, and Graphviz.

25. BoredHumans

- **Publisher**: Impulse Communications, Inc.
- **Size**: 964 instances
- **License**: -
- **Link**: `https://boredhumans.com/prompts.php`
- **Description**: BoredHumans is a diverse and extensive prompt dataset compiled from multiple sources, including Awesome ChatGPT Prompts, Data Science Prompts, and Tree-of-Thought Prompting, among others. Its rich variety covers numerous domains and prompt styles, enabling comprehensive research on prompt engineering, AI model behavior, and in-context learning strategies.

26. CAMEL

- **Publisher**: KAUST
- **Size**: 1659328 instances
- **License**: CC-BY-NC-4.0
- **Link**: `https://huggingface.co/datasets/camel-ai/ai_society`
- **Description**: CAMEL AI Society is a synthetic dialogue corpus comprising 25,000 simulated conversations between GPT-3.5-turbo agents role-playing across 50 distinct user roles and 50 assistant roles on ten tasks per pairing. Available in both chat and instruction formats, each example includes metadata such as role identifiers, original and specified task descriptions, input context, generated responses, and conversation termination reasons. Designed for instruction-tuning and text-generation research, CAMEL is licensed under CC-BY-NC-4.0 and intended solely for non-commercial academic use, acknowledging potential synthetic inaccuracies.

27. ChatGPT & Bing AI Prompts

- **Publisher**: yokoffing
- **Size**: 35 instances
- **License**: CC0-1.0
- **Link**: `https://github.com/yokoffing/ChatGPT-Prompts`
- **Description**: The ChatGPT & Bing AI Prompts dataset offers a diverse collection of prompts designed to optimize interaction with advanced conversational AI models, including ChatGPT and Bing AI. It enables research on prompt engineering techniques, model behavior across different AI platforms, and strategies for enhancing response quality.

28. ChatGPT Data Science Prompts

- **Publisher**: Travis Tang
- **Size**: 60 instances
- **License**: -
- **Link**: `https://github.com/travistangvh/ChatGPT-Data-Science-Prompts`
- **Description**: The ChatGPT Prompts for Data Science dataset offers a curated collection of specialized prompts designed to enhance AI applications in data science tasks. It facilitates research on natural language interfaces for data analysis, model explanation, and automation of complex workflows.

29. ChatGPT Prompts

- **Publisher**: PrathamKumar14
- **Size**: 84 instances
- **License**: -
- **Link**: `https://github.com/PrathamKumar14/ChatGPT-Prompts`
- **Description**: The ChatGPT-Prompts dataset compiles diverse prompt templates focused on educational and productivity applications, including tutoring in web development, algorithm explanation, Excel formulas, social media strategies, and mental health support.

30. ChatGPT Prompts

- **Publisher**: ColorblindAdam
- **Size**: 19 instances
- **License**: -
- **Link**: `https://github.com/ColorblindAdam/ChatGPTPrompts`
- **Description**: The ChatGPT Prompts dataset offers a broad collection of prompts covering diverse topics, designed for use with GPT 3.5. Its value lies in providing versatile, real-world prompt examples that support research on prompt engineering and AI interaction across various domains.

31. ChatGPT Prompts

- **Publisher**: Matheus Nunes Puppe
- **Size**: 36 instances
- **License**: -
- **Link**: `https://github.com/puppe1990/useful_chatgpt_prompts/blob/main/src/promptsData.js`
- **Description**: The ChatGPT Prompts dataset originates from a web application offering a diverse set of prompts generated by OpenAI's GPT-3 model. These prompts serve multiple research purposes, including natural language generation, prompt engineering, and AI-driven creativity.

32. Chinese-DeepSeek-R1-Distill-data-110k

- **Publisher**: Cong Liu et al.
- **Size**: 110K instances
- **License**: Apache-2.0
- **Link**: `https://huggingface.co/datasets/Congliu/Chinese-DeepSeek-R1-Distill-data-110k`
- **Description**: Chinese-DeepSeek-R1-Distill-data-110k is a 110K-entry Chinese dataset distilled from DeepSeek-R1, supporting text generation, text2text generation, and question answering under Apache-2.0. It covers four domains: Math (36 568 samples), Exam (2 432), STEM (12 648) and General (58 352). Each record includes input, reasoning content, output, source repo name and model-assigned score. Data originate from diverse math and instruction corpora, distilled via R1 with temperature 0.6, step-by-step math prompts, and validation using Math-Verify and Qwen2.5-72B.

33. Chinese-DeepSeek-R1-Distill-data-110k-SFT

- **Publisher**: Cong Liu et al.
- **Size**: 110K instances
- **License**: Apache-2.0
- **Link**: `https://huggingface.co/datasets/Congliu/Chinese-DeepSeek-R1-Distill-data-110k-SFT`
- **Description**: Licensed under Apache-2.0, Chinese-DeepSeek-R1-Distill-data-110k-SFT is an open-source, Chinese-language instruction-tuning dataset distilled from DeepSeek-R1 outputs, formatted for direct supervised fine-tuning. It comprises 110K examples spanning math (36.6K), exam questions (2.4K), STEM (12.6K), and diverse general prompts (58.4K). Prompts are sourced from multiple Chinese math and STEM repositories, with distillation performed at temperature 0.6 and special step-by-step cues for calculations. Each sample includes integrated reasoning, answers, and model-based scores, facilitating reproducibility of high-performance SFT training. It supports text-generation, text-to-text generation, and question-answering tasks.

34. CoCoNot

- **Publisher**: Allen Institute for AI et al.
- **Size**: 13784 instances
- **License**: ODC-BY-1.0
- **Link**: https://huggingface.co/datasets/allenai/coconot
- **Description**: CoCoNot is a novel English dataset for benchmarking and improving contextual noncompliance in chat-based language models. It offers three configurations: "original" contains 11K training and 1K test examples of user prompts that models should refuse; "contrast" comprises 379 test examples requiring compliant responses; and "pref" holds 927 preference-labeled training pairs contrasting optimal with noncompliant replies. Examples include metadata (id, category, subcategory, prompt, response) across five noncompliance categories. Developed by AI2, CoCoNot supports text-generation tasks aimed at refining models' refusal behavior.

35. COIG-CQIA

- **Publisher**: Shenzhen Institute of Advanced Technology et al.
- **Size**: 44694 instances
- **License**: -
- **Link**: https://huggingface.co/datasets/m-a-p/COIG-CQIA
- **Description**: COIG-CQIA (Chinese Open Instruction Generalist - Quality is All You Need) is a high-quality, open-source Chinese instruction tuning dataset designed to align language models with human interactive behavior. It aggregates over 45,000 manually cleansed, restructured, and reviewed examples spanning social media dialogs, encyclopedic articles, exam questions, finance, medical, legal, traditional culture, and NLP tasks. Each entry includes instruction, optional input, output, task type, domain, and human verification metadata. COIG-CQIA aims to facilitate instruction fine-tuning for Chinese NLP research and applications.

36. CVQA

- **Publisher**: MBZUAI
- **Size**: 10374 instances
- **License**: Mixed
- **Link**: https://huggingface.co/datasets/afaji/cvqa
- **Description**: CVQA is a culturally diverse, multilingual visual question-answering benchmark featuring over 10,000 image-based questions across 39 country-language pairs. Each sample includes a locally posed query, its English translation, four answer options in both languages, and metadata such as image source, license, category, and a unique ID. Questions span ten thematic categories and images originate from self-contributed and external sources under various licenses. Designed primarily as a test set, CVQA facilitates evaluation of VQA models on nuanced, culturally contextualized visual understanding.

37. databricks-dolly-15K

- **Publisher**: Databricks
- **Size**: 15011 instances
- **License**: CC-BY-SA-3.0
- **Link**: https://huggingface.co/datasets/databricks/databricks-dolly-15k
- **Description**: Databricks-dolly-15K is an open-source corpus of over 15,000 human-generated instruction-response pairs created by Databricks employees across eight behavioral categories defined by InstructGPT, including brainstorming, classification, closed and open QA, generation, information extraction, and summarization. Provided under a CC-BY-SA 3.0 license, this English-language dataset supports academic or commercial use. With context passages drawn from Wikipedia when required, it enables training and fine-tuning of large language models, as well as synthetic data generation and data augmentation for robust, scalable instruction-following capabilities.

38. DeepMath-103K

- **Publisher**: Tencent et al.
- **Size**: 103110 instances
- **License**: MIT
- **Link**: `https://huggingface.co/datasets/zwhe99/DeepMath-103K`
- **Description**: DeepMath-103K is a large-scale, MIT-licensed dataset comprising 103K challenging mathematical problems tailored for text-to-text and text-generation tasks. Each example includes a problem statement, a hierarchically classified topic, a numerical difficulty score, three distinct reasoning pathways (R1 solutions), and a verifiable final answer. Designed to support reinforcement learning and supervised fine-tuning, it enables difficulty-aware training, topic-specific evaluation, and robust rule-based reward shaping. Sourced and decontaminated to minimize test leakage, DeepMath-103K drives advances in automated mathematical reasoning research and diverse research areas.

39. DeepSeek-Prover-V1
    - **Publisher**: DeepSeek
    - **Size**: 27503 instances
    - **License**: deepseek-license
    - **Link**: `https://huggingface.co/datasets/deepseek-ai/DeepSeek-Prover-V1`
    - **Description**: DeepSeek-Prover-V1 is a large-scale synthetic proof dataset for Lean 4 theorem proving. It comprises 8 million formal statements and corresponding proofs generated from high-school and undergraduate-level mathematical contest problems. Natural language problems are translated into formal Lean 4 statements, filtered for quality, and paired with automatically generated proofs. Released under the deepseek-license, this dataset enables fine-tuning of large language models, improving whole-proof generation accuracy on benchmarks like miniF2F and FIMO. It supports research in formalized mathematical reasoning, automated theorem proving.

40. DialogStudio
    - **Publisher**: Salesforce AI et al.
    - **Size**: 87 datasets
    - **License**: Apache-2.0
    - **Link**: `https://huggingface.co/datasets/Salesforce/dialogstudio`
    - **Description**: DialogStudio is a large-scale, unified collection of dialogue datasets curated to advance conversational AI. It integrates a wide range of domains—such as task-oriented dialogue, open-domain conversation, knowledge-grounded dialogue, and more—while preserving original metadata and structure. The dataset supports instruction-tuned training and evaluation across over 30 datasets with consistency. It includes model checkpoints (e.g., dialogstudio-t5-base-v1.0) and evaluation scripts using GPT-3.5 for quality metrics like coherence, completeness, and correctness. DialogStudio serves as a robust benchmark for multi-task generalization, instruction-following, and multi-domain dialogue modeling.

41. DMind_Benchmark
    - **Publisher**: Zhejiang Univerisity et al.
    - **Size**: 1869 instances
    - **License**: -
    - **Link**: `https://huggingface.co/datasets/DMindAI/DMind_Benchmark`
    - **Description**: DMind_Benchmark is a comprehensive dataset for evaluating large language models on blockchain, cryptocurrency, and Web3 knowledge. It provides objective (multiple choice) and subjective (open ended) questions across nine domains: Fundamentals, Infrastructure, Smart Contracts, DeFi, DAOs, NFTs, Security, Tokenomics, and MEME coins—organized into CSV and JSONL splits. The benchmark supports diverse question types—calculations, code audits, risk and scenario analyses—with automated scoring and evaluation. It features standardized data configurations, leaderboards, and extensible evaluation pipelines for comparative analysis of LLM performance in specialized Web3 tasks.

42. Dynosaur
    - **Publisher**: UCLA et al.

- **Size**: 801900 instances
- **License**: Apache-2.0
- **Link**: `https://huggingface.co/datasets/Dynosaur/dynosaur-sub-superni`
- **Description**: Dynosaur introduces a dynamic and low-cost paradigm for curating instruction-tuning datasets. It automatically generates diverse instructions by leveraging metadata from HuggingFace datasets, combined with LLM-based instruction synthesis (e.g., via ChatGPT). The result is Dynosaur-full, a large-scale dataset (800K+ samples, generated at \$11.5) that supports dynamic growth and general-purpose instruction-tuning. Empirically, models fine-tuned on Dynosaur outperform Alpaca and GPT-4-Instruct baselines on Super-NI. The project includes: metadata crawling tools, instruction generation pipelines, and fine-tuned T5-3B and LLaMA-7B models. All generated instructions are under Apache 2.0, with task data adhering to original dataset licenses.

43. Exploring the Possibilities of AI Prompts Over 200 Ideas
    - **Publisher**: Muhammad Bilal
    - **Size**: 165 instances
    - **License**: MIT
    - **Link**: `https://github.com/bilalnawaz072/AI-Prompts-200-Ideas`
    - **Description**: "Exploring the Possibilities of AI Prompts Over 200 Ideas" is a comprehensive dataset featuring over 200 prompts spanning diverse marketing and content creation domains such as blog writing, email marketing, social media ads, influencer campaigns, and copywriting.

44. Firefly
    - **Publisher**: YeungNLP
    - **Size**: 1649399 instances
    - **License**: -
    - **Link**: `https://huggingface.co/datasets/YeungNLP/firefly-train-1.1M`
    - **Description**: Firefly is a Chinese instruction-tuning dataset comprising 1.15 million high-quality examples drawn from 23 common Chinese natural language processing datasets. Each example includes a task type, an input prompt, and a target output, ensuring diverse coverage. Data templates were manually designed for each task to ensure quality and richness. Token length analysis shows that most examples are under 600 tokens. Firefly was used to train the Firefly-1b4 Chinese dialogue LLM, available on GitHub and Hugging Face, fostering reproducibility, community collaboration.

45. Flan 2021
    - **Publisher**: Google Research
    - **Size**: 62 datasets
    - **License**: Apache-2.0
    - **Link**: `https://github.com/google-research/FLAN`
    - **Description**: The FLAN Instruction Tuning Repository provides datasets and code to generate instruction tuning collections that improve language model generalization and zero-shot performance. Originating with FLAN 2021 and expanded in the FLAN Collection, this resource supports research on fine-tuning methods that enable large models to better follow human instructions. It underpins influential models like FLAN-T5 and FLAN-PaLM, facilitating advances in instruction-based learning and enabling systematic exploration of tuning strategies for enhanced natural language understanding.

46. Flan 2022
    - **Publisher**: Google Research
    - **Size**: 1836 datasets
    - **License**: Apache-2.0
    - **Link**: `https://huggingface.co/datasets/SirNeural/flan_v2`

- **Description**: This dataset aggregates tasks from Flan, T0, Super-Natural Instructions, Chain-of-Thought, and Dialog into a training split. Each task is provided in zero-/few-shot and option/no-option formats as JSONL entries including inputs, targets, and task identifiers. Released under Apache-2.0, it includes scripts for building dependencies, fixing version mismatches, and exporting per-task JSONL data. Mixing ratios can be tuned for optimal downstream performance via guidelines in the associated paper and public GitHub repository.

47. Flan-mini
    - **Publisher**: Singapore University of Technology and Design
    - **Size**: 1.34M instances
    - **License**: CC
    - **Link**: `https://huggingface.co/datasets/declare-lab/flan-mini`
    - **Description**: Flan-mini is a curated 1.34 M-example subset of the FLAN instruction-tuning collection augmented with code and conversational tasks. It pools 388K Flan2021 instructions, 320K public prompt templates, 200K Natural Instructions v2 instances, 100K chain-of-thought examples, plus code datasets (100K Code Search, 50K Code Contests, 50K APPS). It further integrates 132K ChatGPT-generated examples from GPT-4-Alpaca, Code-Alpaca, and ShareGPT. Each example is randomly paired with handcrafted prompt templates for zero- or few-shot fine-tuning, ensuring diverse task coverage. Released under a permissive CC license.

48. GEdit-Bench
    - **Publisher**: StepFun
    - **Size**: 1212 instances
    - **License**: MIT
    - **Link**: `https://huggingface.co/datasets/stepfun-ai/GEdit-Bench`
    - **Description**: GEdit-Bench is a novel benchmark dataset designed to facilitate authentic evaluation of general-purpose image editing models. Developed alongside the Step1X-Edit framework, it emphasizes real-world usage scenarios and supports a diverse array of image-to-image editing tasks. Offered under the MIT license, GEdit-Bench provides a standardized testbed for assessing algorithmic performance, robustness, and versatility in scalable practical editing workflows.

49. GPT4All
    - **Publisher**: nomic-ai
    - **Size**: 739259 instances
    - **License**: MIT
    - **Link**: `https://huggingface.co/datasets/nomic-ai/gpt4all_prompt_generations`
    - **Description**: The GPT4All dataset comprises 437,604 English prompt-response pairs drawn from diverse sources to facilitate training and fine-tuning of open-source text generation models. It pairs user prompts with AI-generated replies and source metadata, covering various topics and styles. Released under Apache-2.0, the training split occupies approximately 782 MB on disk and requires 398 MB download. Curated by Nomic AI, GPT4All supports reproducible research in conversational AI. Hosted on GitHub with an accompanying technical report. It includes benchmarks along with extensive tests.

50. GraphWalks
    - **Publisher**: OpenAI
    - **Size**: 1150 instances
    - **License**: MIT
    - **Link**: `https://huggingface.co/datasets/openai/graphwalks`
    - **Description**: GraphWalks is an open-source benchmark dataset designed to evaluate multi-hop reasoning over long graph contexts. Released under the MIT license, it provides directed graphs as edge lists alongside user-specified operations—such as breadth-first searches or parent retrieval—for models to execute. Each prompt comprises three demonstration examples, a target graph, and a query, with expected outputs formatted as node ID lists. Accompanying metadata includes prompt character counts and problem types. Standardized extraction and F1-based grading scripts ensure consistent answer parsing and evaluation.

51. GSM8K

- **Publisher**: OpenAI
- **Size**: 17584 instances
- **License**: MIT
- **Link**: `https://huggingface.co/datasets/openai/gsm8k`
- **Description**: GSM8K (Grade School Math 8K) is an English monolingual dataset of 8.8K crowd-sourced grade school math word problems paired with multi-step solutions. It contains a main configuration and a Socratic variant, each offering questions and answers with calculator annotations and step-by-step reasoning expressed in natural language. Problems require two to eight elementary arithmetic steps. Split into training (7,473 examples) and test (1,319 examples), GSM8K supports text-to-text generation benchmarks under MIT license. All annotations were crowdsourced via Upwork and Surge AI.

52. HARDMath

- **Publisher**: Harvard University
- **Size**: 1060 instances
- **License**: MIT
- **Link**: `https://github.com/sarahmart/HARDMath`
- **Description**: HARDMath is a benchmark dataset designed to evaluate advanced mathematical reasoning in large language models, focusing on challenging graduate-level applied mathematics problems. Unlike existing benchmarks that emphasize straightforward undergraduate problems, HARDMath includes complex problems requiring approximation techniques, mathematical intuition, and sophisticated problem-solving. It contains over 1,000 diverse problems across multiple categories, including a special set of handwritten word problems demanding asymptotic reasoning in realistic contexts. HARDMath thus fills a critical gap for rigorous evaluation of mathematical capabilities in AI research.

53. HC3

- **Publisher**: SimpleAI
- **Size**: 37175 instances
- **License**: CC-BY-SA-4.0
- **Link**: `https://huggingface.co/datasets/Hello-SimpleAI/HC3`
- **Description**: The Human ChatGPT Comparison Corpus (HC3) is the first large-scale bilingual dataset enabling direct comparison of human and ChatGPT-generated text. Spanning English and Chinese samples, it encompasses between 10,000 and 100,000 prompt-response pairs covering tasks such as text classification, question-answering, sentence similarity, and zero-shot classification. Released under a CC-BY-SA license, HC3 supports research in performance evaluation, detection, and analysis of AI-generated content. Accompanying code, models, and benchmarks are available on GitHub, facilitating open science, reproducible experimentation, and collaborative, community-driven global efforts.

54. hh-rlhf

- **Publisher**: Anthropic
- **Size**: 14M instances
- **License**: MIT
- **Link**: `https://github.com/anthropics/hh-rlhf`
- **Description**: hh-rlhf provides valuable human preference data focused on helpfulness and harmlessness for training safer AI assistants using Reinforcement Learning from Human Feedback. It includes paired comparison data from base and iterated models, as well as red teaming transcripts designed to expose model vulnerabilities.

55. InstructDial

- **Publisher**: Carnegie Mellon University
- **Size**: 59 datasets
- **License**: Apache-2.0
- **Link**: `https://github.com/prakharguptaz/Instructdial`

- **Description**: InstructDial is a comprehensive instruction tuning framework designed to improve zero-shot and few-shot generalization in dialogue systems. It unifies 48 diverse dialogue tasks from 59 datasets into a text-to-text format, enabling models to learn across multiple dialogue-related functions such as understanding, generation, and intent detection.

56. InstructionWild_v1

- **Publisher**: National University of Singapore
- **Size**: 104K instances
- **License**: Non-Commercial Research Purpose
- **Link**: `https://github.com/XueFuzhao/InstructionWild`
- **Description**: InstructWild is a large-scale, user-sourced instruction dataset comprising over 110K high-quality, diverse instructions collected from real ChatGPT usage shared on social media. Unlike previous synthetic datasets, InstructWild emphasizes authentic, varied user intents without relying on self-generated instructions. It supports both English and Chinese and enhances model capabilities in generation, open-domain QA, and creative thinking. This dataset provides a valuable resource for instruction tuning, advancing large language model generalization with naturally occurring user prompts.

57. InstructionWild_v2

- **Publisher**: National University of Singapore
- **Size**: 110K instances
- **License**: Non-Commercial Research Purpose
- **Link**: `https://github.com/XueFuzhao/InstructionWild`

58. Intellect-2-RL-Dataset

- **Publisher**: PrimeIntellect
- **Size**: 284741 instances
- **License**: Apache-2.0
- **Link**: `https://huggingface.co/datasets/PrimeIntellect/INTELLECT-2-RL-Dataset`
- **Description**: Intellect-2-RL-Dataset is a large-scale collection of 284,741 training examples, designed for reinforcement learning in mathematical and coding problem solving. Each entry includes a unique problem_id, a task_type label, the problem prompt, verification_info detailing solution validity, and a baseline solve_rate from the Qwen-R1-Distill-7B model. Released under Apache-2.0 license, this dataset supports fine-tuning and evaluation of reasoning-oriented language models, facilitating research on algorithmic proficiency and reward-driven optimization within distributed asynchronous RL frameworks.

59. LaMini-instruction

- **Publisher**: Monash University et al.
- **Size**: 2585615 instances
- **License**: CC-BY-NC-4.0
- **Link**: `https://huggingface.co/datasets/MBZUAI/LaMini-instruction`
- **Description**: LaMini-Instruction is an English text-to-text generation dataset comprising 2.58M instruction-response pairs distilled from GPT-3.5-Turbo. Each sample includes an instruction, a corresponding model-generated response, and the instruction's provenance— drawn from sources such as Alpaca, FLAN, P3, and Self-Instruct. Released under CC-BY-NC 4.0, it spans a single training split of over 1.16 GB and supports fine-tuning of compact language models. LaMini-Instruction enables research in instruction-based learning but inherits biases and errors from its GPT-3.5 teacher.

60. LCCC

- **Publisher**: Tsinghua University et al.
- **Size**: 12M instances
- **License**: MIT
- **Link**: `https://huggingface.co/datasets/thu-coai/lccc`

- **Description**: LCCC (Large-scale Cleaned Chinese Conversation Corpus) is a monolingual Chinese dialogue dataset with over 12 million conversations collected from social media. A strict and rigorous cleaning pipeline—including manual rules and classifier-based filters—removes noisy utterances such as offensive language, emojis, special symbols, ungrammatical or incoherent exchanges. The base configuration offers 6.8 M training samples with 20 K validation and 10 K test dialogues, while a larger variant provides 12 M training instances. Licensed under MIT, LCCC supports two key tasks: response generation and retrieval.

61. LIMA-sft

    - **Publisher**: Meta AI et al.
    - **Size**: 1330 instances
    - **License**: CC-BY-NC-SA
    - **Link**: https://huggingface.co/datasets/GAIR/lima
    - **Description**: The LIMA dataset contains 1,000 high-quality prompt-response pairs designed to align language models with the style of a helpful AI assistant. Prompts are diverse, sourced from Stack Exchange, wikiHow, WritingPrompts, Natural Instructions, and manually authored examples. Despite limited size ( 750K tokens), all responses are stylistically consistent. The dataset includes a 50-example development set and a 300-prompt test set. LIMA demonstrates that small, curated datasets can be highly effective for instruction tuning and alignment of pretrained language models.

62. Llama-Nemotron-Post-Training-Dataset

    - **Publisher**: NVIDIA
    - **Size**: 33011757 instances
    - **License**: CC-BY-4.0
    - **Link**: https://huggingface.co/datasets/nvidia/Llama-Nemotron-Post-Training-Dataset
    - **Description**: The Llama-Nemotron-Post-Training-Dataset is a comprehensive dataset of synthetic SFT and RL samples designed to bolster reasoning, code, math, science, chat, and safety capabilities for NVIDIA's Llama-3 Nemotron series. It includes over 33M SFT examples across code, math, science, chat, and safety, plus 56K instruction-following RL examples. Data is sourced from public corpora or synthetically generated, filtered for quality and complexity. Released under CC-BY-4.0, it supports training and evaluation of efficient open-source LLMs offering a flexible accuracy-efficiency tradeoff and transparent development.

63. LMSYS-Chat-1M

    - **Publisher**: UC Berkeley et al.
    - **Size**: 1M instances
    - **License**: LMSYS-Chat-1M license
    - **Link**: https://huggingface.co/datasets/lmsys/lmsys-chat-1m
    - **Description**: LMSYS-Chat-1M is a large-scale dataset of one million real-world LLM conversations, collected from 210K users interacting with 25 models via Chatbot Arena and Vicuna demo (April-August 2023). Each conversation includes model metadata, OpenAI-style JSON formatting, language tags, and moderation labels. Personally identifiable information is redacted. This dataset enables research on LLM alignment, safety, evaluation, and user behavior in the wild, offering unique insights into real-world usage patterns and content moderation challenges in multi-model deployment scenarios.

64. LongForm

    - **Publisher**: LMU Munich et al.
    - **Size**: 27739 instances
    - **License**: MIT
    - **Link**: https://huggingface.co/datasets/akoksal/LongForm
    - **Description**: LongForm is a 27K-example English instruction-following dataset under MIT license, for tasks like table QA, summarization, text generation, question answering. It collects human-written documents from C4 (10K) and Wikipedia (5K), reverse-engineered

instructions via LLMs, and structured sources including Stack Exchange (4.4K) and Wiki-How (2.5K). It also covers QA, email writing, grammar correction, story/poem generation and summarization from NIv2, Big Bench, BEA-GEC, Enron. Split into 23.6K train, 2K validation and 2K test, it supports instruction tuning and is publicly available.

65. Math_CoT_Arabic_English_Reasoning

- **Publisher**: Miscovery AI
- **Size**: 2834 instances
- **License**: MIT
- **Link**: https://huggingface.co/datasets/miscovery/Math_CoT_Arabic_English_Reasoning
- **Description**: Math CoT Arabic English Reasoning is a bilingual dataset of 1K-10K meticu-lously curated English and Arabic math problems with explicit chain-of-thought solutions. Spanning 21 categories from arithmetic to topology and logic, it offers human-verified, step-by-step reasoning examples in parallel languages. Structured in JSON with questions, answers, comprehensive metadata, category labels, and word counts, it supports question-answering, text generation, and mask-filling benchmarks. Licensed under MIT, it's ideal for robust multilingual mathematical reasoning research, cross-lingual model evaluation, and educational AI assistant development.

66. medical-o1-reasoning-SFT

- **Publisher**: The Chinese University of Hong Kong, Shenzhen et al.
- **Size**: 90120 instances
- **License**: Apache-2.0
- **Link**: https://huggingface.co/datasets/FreedomIntelligence/medical-o1-reasoning-SFT
- **Description**: medical-o1-reasoning-SFT is a supervised fine-tuning dataset designed to enhance advanced medical reasoning in HuatuoGPT-o1. It comprises English and Chinese instruction-response pairs generated by GPT-4o on verifiable clinical problems, validated by a medical verifier. Released under an Apache-2.0 license, the dataset supports ques-tion answering and text generation, offering separate configurations for monolingual and mixed-language data. It aims to refine model performance on complex biomedical tasks by leveraging rigorous problem-solving chains, with full details available in the accompanying paper and GitHub repository.

67. medical-o1-verifiable-problem

- **Publisher**: The Chinese University of Hong Kong, Shenzhen et al.
- **Size**: 40644 instances
- **License**: Apache-2.0
- **Link**: https://huggingface.co/datasets/FreedomIntelligence/medical-o1-verifiable-problem
- **Description**: medical-o1-verifiable-problem is an Apache-2.0 licensed dataset comprising open-ended medical reasoning problems designed to improve large language models' diag-nostic and procedural knowledge. It supports question-answering and text-generation tasks, presenting each instance as a challenging exam-style prompt paired with a verifiable, expert-derived answer. Published in English under a single default configuration with training data provided in JSON format, it allows systematic evaluation and refinement of LLM outputs.

68. Medical-R1-Distill-Data

- **Publisher**: The Chinese University of Hong Kong, Shenzhen et al.
- **Size**: 22000 instances
- **License**: Apache-2.0
- **Link**: https://huggingface.co/datasets/FreedomIntelligence/Medical-R1-Distill-Data
- **Description**: Medical-R1-Distill-Data is an Apache-2.0 licensed instruction fine-tuning dataset distilled from Deepseek-R1's Full Power Version using medical verifiable problems sourced from HuatuoGPT-o1. It supports English and Chinese, and is tailored for question-answering and text-generation tasks in medical and biology domains. The dataset captures

reasoning chains from the native Deepseek-R1 API, facilitating model initialization with robust medical reasoning. A Chinese counterpart is available separately. Methodology and guidelines are provided in the associated paper and GitHub repository. It comprises SFT examples from medical_r1_distill_sft.json.

69. MedReason

- **Publisher**: UC Santa Cruz et al.
- **Size**: 32682 instances
- **License**: Apache-2.0
- **Link**: `https://huggingface.co/datasets/UCSC-VLAA/MedReason`
- **Description**: MedReason is a large-scale medical reasoning dataset combining seven clinical question-answer sources with a structured knowledge graph to produce detailed chains of reasoning. It contains 32,682 QA pairs, each annotated with step-by-step explanatory "thinking paths" derived from standardized medical KG relations. Designed to enhance the faithfulness and interpretability of medical problem-solving in large language models, MedReason enables fine-tuning of models such as MedReason-8B, which demonstrates state-of-the-art performance. Released under Apache-2.0, this open-source dataset aims to foster transparent medical QA systems.

70. Medtrinity-25M

- **Publisher**: Huazhong University of Science and Technology et al.
- **Size**: 24922190 instances
- **License**: Mixed
- **Link**: `https://huggingface.co/datasets/UCSC-VLAA/MedTrinity-25M`
- **Description**: MedTrinity-25M is a large-scale multimodal medical dataset featuring over 25 million images from 10 imaging modalities. It provides multigranular annotations for 65+ diseases, including textual descriptions, bounding boxes, segmentation masks, and inter-region relationships. Supporting both vision-centric and multimodal tasks like classification, segmentation, and report generation, it facilitates large-scale pretraining for medical foundation models. Public access includes an 18M image-text pair subset. The dataset is organized in shards with structured metadata for scalable research and development.

71. MMInstruct-GPT4V

- **Publisher**: Shanghai AI Laboratory et al.
- **Size**: 378186 instances
- **License**: Apache-2.0
- **Link**: `https://huggingface.co/datasets/yuecao0119/MMInstruct-GPT4V`
- **Description**: MMInstruct-GPT4V is a multilingual multi-modal instruction tuning dataset for visual question answering and image captioning, licensed under Apache-2.0. It comprises three configurations—qa_en, caption_en, and caption_cn—covering English QA ( 216K examples), English captions ( 18K examples), and Chinese captions ( 144K examples) in JSONL train splits. Total size ranges between 100K and 1M instances. Designed to leverage GPT-4V for high-quality instruction generation, it supports both one-shot and multi-round interactions, enabling robust supervised fine-tuning of vision-language models targeting visual-question-answering and question-answering tasks with enhanced robustness.

72. Mol-Instructions

- **Publisher**: Zhejiang University
- **Size**: over 2 million instances
- **License**: CC-BY-4.0
- **Link**: `https://huggingface.co/datasets/zjunlp/Mol-Instructions`
- **Description**: Mol-Instructions is an open-access, large-scale biomolecular instruction dataset with 100M-1B examples designed to facilitate instruction-tuning of large language models on chemistry and biology tasks. Comprised of three core components—148.4K molecule-oriented instructions (e.g. reaction prediction, property prediction), 505K protein-oriented instructions (e.g. structure/function prediction, protein design) and 53K biomolecular text

instructions (e.g. chemical entity recognition, QA)—it supports diverse molecule, protein and NLP tasks. Released under CC-BY-4.0 on Hugging Face, Mol-Instructions aims to advance biomolecular AI research.

73. MOSS_002_sft_data

- **Publisher**: Fudan University
- **Size**: 1161137 instances
- **License**: CC-BY-NC-4.0
- **Link**: `https://huggingface.co/datasets/fnlp/moss-002-sft-data`
- **Description**: MOSS_002_sft_data is an open-source bilingual conversational dataset designed for fine-tuning MOSS-002. It encompasses over one million samples in English and Chinese across five splits—helpfulness, honesty and harmlessness—totaling 2.16 GB of text. User prompts are expanded from human-written seeds via a Self-Instruct-style pipeline, while model responses are synthesized with text-davinci-003. Harmlessness examples in English leverage Anthropic's red-teaming attempts. Licensed under CC-BY-4.0, the resource supports text-generation and conversational modeling research within the 1-10 M size category. It is accessible via GitHub and homepage.

74. MRCR

- **Publisher**: OpenAI
- **Size**: 2400 instances
- **License**: MIT
- **Link**: `https://huggingface.co/datasets/openai/mrcr`
- **Description**: OpenAI MRCR (Multi-round co-reference resolution) is a long-context benchmark evaluating LLMs' ability to find multiple identical requests ("needles") hidden within multi-turn conversations. Inspired by Gemini's MRCR, it embeds 2, 4, or 8 duplicate prompts (e.g., "Write a poem about tapirs") among distractors, prompting models to retrieve the i-th instance. It comprises 438 entities, 10 writing formats, and 100 samples per bin across eight token-based bins up to one million tokens. Evaluation uses SequenceMatcher ratio and mandates an alphanumeric hash prefix.

75. NATURAL INSTRUCTIONS

- **Publisher**: Allen Institute for AI et al.
- **Size**: 61 datasets
- **License**: Apache-2.0
- **Link**: `https://huggingface.co/datasets/Muennighoff/natural-instructions`
- **Description**: NATURAL INSTRUCTIONS is a monolingual English dataset derived from Super-Natural-Instructions, offering 1,600+ NLP tasks for training, validation, and testing. Size ranges between 100 million and one billion examples. Curated by crowdsourced and expert annotators, it covers classification, generation, and reasoning across reading comprehension, commonsense, summarization, arithmetic, logic, and dialog. With over 100 M examples, it provides diverse input-output mappings while enabling deduplication by unique IDs or input fields. Tasks span question answering, text modification, summarization, and beyond, supporting robust instruction-following model development.

76. Nemotron-CrossThink

- **Publisher**: NVIDIA
- **Size**: 588645 instances
- **License**: CC-BY-4.0
- **Link**: `https://huggingface.co/datasets/nvidia/Nemotron-CrossThink`
- **Description**: Nemotron-CrossThink is a multi-domain reinforcement learning dataset designed to enhance both general-purpose and mathematical reasoning in large language models. It comprises two subsets: Nemotron-CrossThink-QA with high-quality question-answer pairs across STEM, humanities, and sciences, and Nemotron-CrossThink-Math featuring persona-driven, multi-step math problems. Data is curated from CommonCrawl and open-source

books, standardized via structured templates into multiple-choice and open-ended formats, filtered for verifiability, and used to train RL policies with Group Relative Policy Optimization. Licensed under CC-BY-4.0, it supports AI development.

77. New Yorker Caption Ranking

- **Publisher**: University of Wisconsin-Madison et al.
- **Size**: 2183522 instances
- **License**: CC-BY-NC-4.0
- **Link**: `https://huggingface.co/datasets/yguooo/newyorker_caption_ranking`
- **Description**: The New Yorker Caption Ranking dataset comprises over 250 million massive crowdsourced humor ratings on more than 2.2 million captions collected from eight years of New Yorker cartoon caption contests. Structured into description, ranking, and cartoon subsets, it provides multimodal inputs paired with human preference judgments for training and evaluating creative text-generation models. The dataset supports rigorous benchmark development using human and GPT-4 assessments, showing current fine-tuning methods underperform top human contestants. Licensed under CC-BY-NC-4.0 and accessible via Hugging Face.

78. No Robots

- **Publisher**: Hugging Face H4
- **Size**: 10000 instances
- **License**: CC-BY-NC-4.0
- **Link**: `https://huggingface.co/datasets/HuggingFaceH4/no_robots`
- **Description**: No Robots is a high-quality, human-curated instruction dataset comprising 10,000 examples for supervised fine-tuning of language models. It includes 9,500 training and 500 test instances across ten single-turn categories—Generation, Open QA, Brainstorm, Chat, Rewrite, Summarize, Coding, Classify, Closed QA, and Extract—totaling roughly 17 MB of English text under CC-BY-NC-4.0. Each example consists of a prompt with unique ID, structured message history (system, user, assistant), and category labels. It enables models to learn diverse instruction-following behaviors and robustly supports reproducibility.

79. NuminaMath-1.5

- **Publisher**: Numina
- **Size**: 896215 instances
- **License**: Apache-2.0
- **Link**: `https://huggingface.co/datasets/AI-MO/NuminaMath-1.5`
- **Description**: NuminaMath-1.5 is an open-source, large-scale post-training dataset comprising about 900 000 competition-level mathematics problems paired with chain-of-thought solutions. It covers diverse sources—from Chinese high school exams to US and international Olympiads—and spans domains like algebra, geometry, number theory, combinatorics, calculus, and puzzles. Each entry includes metadata fields (answer, problem_type, question_type) for verifiable outputs. Recent additions feature manually verified Olympiad references and curated contest data while synthetic problems were removed. Licensed under Apache 2.0, NuminaMath-1.5 supports advanced text-generation research in mathematical reasoning.

80. OASST1

- **Publisher**: OpenAssistant
- **Size**: 161443 instances
- **License**: Apache-2.0
- **Link**: `https://huggingface.co/datasets/OpenAssistant/oasst1`
- **Description**: OpenAssistant Conversations (OASST1) is a human-generated, human-annotated corpus with 161,443 messages in 66,497 conversation trees across 35 languages. It includes over 461,000 quality ratings and more than 10,000 fully annotated trees. Each record contains metadata (IDs, timestamps), conversational structure (parent and tree IDs), role and language labels, toxicity and quality scores, emoji labels. Data comes in nested JSONL or flat parquet via HuggingFace, with 84,437 training and 4,401 validation splits, supporting supervised fine-tuning and reward model development. Licensed under Apache-2.0.

81. OIG

- **Publisher**: LAION
- **Size**: 3878622 instances
- **License**: Apache-2.0
- **Link**: `https://huggingface.co/datasets/laion/OIG`
- **Description**: Open Instruction Generalist (OIG) is a large-scale instruction-tuning dataset released under Apache-2.0 license. It comprises 44 million JSONL entries pairing human instructions with model responses for continued pretraining, accompanied by a smaller high-quality subset (OIG-small-chip2) optimized for finetuning. OIG unifies diverse sources—ranging from Wikipedia dialogs, math problems, and code examples to summarization and question-answering corpora—into a consistent format. Designed to transform pretrained models into instruction-following agents, it supports scalable development of helpful language systems and targets one trillion tokens of instructions.

82. OL-CC

- **Publisher**: BAAI
- **Size**: 11655 instances
- **License**: Apache-2.0
- **Link**: `https://huggingface.co/datasets/lorinma/BAAI_OL-CC`
- **Description**: OL-CC is the first open source Chinese conversational instruction dataset collected via crowdsourcing on OpenLabel. It includes 10,006 instruction-answer pairs and 1,649 standalone instructions across tasks such as question-answering, text generation, extraction, rewriting, classification, brainstorming, chit-chat, logic and math. A total of 276 volunteers alternately played user and AI assistant roles to produce the data. Licensed under Apache-2.0 and sized between 10K and 100K examples, OL-CC offers rich, human-generated Chinese instructional dialogues for AI research.

83. OpenCodeInstruct

- **Publisher**: NVIDIA
- **Size**: 5M instances
- **License**: CC-BY-4.0
- **Link**: `https://huggingface.co/datasets/nvidia/OpenCodeInstruct`
- **Description**: OpenCodeInstruct is a large-scale open-access instruction tuning dataset for code language models provided under the CC-BY-4.0 license. It comprises five million examples across generic and algorithmic coding tasks, with fields including id, input, output, domain, generation_algorithm, llm_judgement, unit_tests, tests_execution_status, and average_test_score. It supports supervised fine-tuning of code models and is accessible via the HuggingFace datasets library. Developed by NVIDIA for research and use, it accelerates code generation benchmarks and model evaluation.

84. OpenCodeReasoning

- **Publisher**: NVIDIA
- **Size**: 735255 instances
- **License**: CC-BY-4.0
- **Link**: `https://huggingface.co/datasets/nvidia/OpenCodeReasoning`
- **Description**: OpenCodeReasoning is a large-scale synthetic dataset designed to distill reasoning capabilities for Python-based competitive programming. It comprises 735,255 samples covering 28,319 unique problems sourced from platforms like CodeForces, AtCoder, and LeetCode. The dataset features two configurations: split_0 includes full problem statements and model responses, while split_1 references external datasets via index placeholders. Each example contains identifiers, source metadata, difficulty labels, and code solutions. Licensed under CC-BY-4.0, OpenCodeReasoning supports supervised fine-tuning of language models for code generation tasks.

85. OpenMathReasoning

- **Publisher**: NVIDIA
- **Size**: 5469691 instances

- **License**: CC-BY-4.0
- **Link**: `https://huggingface.co/datasets/nvidia/OpenMathReasoning`
- **Description**: OpenMathReasoning is a large-scale English math-reasoning dataset (cc-by-4.0) comprising 290K+ olympiad problems with 3.2M chain-of-thought (CoT), 1.7M tool-integrated reasoning (TIR), and 566K GenSelect solution samples. Sourced from AoPS and processed with Qwen2.5-32B, DeepSeek-R1, and QwQ-32B, each record includes problem statements, generated solutions, expected answers, inference modes, metadata, and pass-rate metrics. Available in cot, tir, and genselect splits, it underpins state-of-the-art LLM training and evaluation in question-answering and text-generation.

86. OpenOrca

- **Publisher**: Microsoft Research
- **Size**: 4233923 instances
- **License**: MIT
- **Link**: `https://huggingface.co/datasets/Open-Orca/OpenOrca`
- **Description**: OpenOrca is an open English dataset licensed under MIT that augments the FLAN Collection with over 4 million GPT-3.5 and GPT-4 responses. It provides system prompts, questions, and AI-generated answers with detailed reasoning traces in tabular format. Tailored for a wide range of tasks including conversational modeling, classification, summarization, question-answering, and zero-shot scenarios. OpenOrca facilitates instruction tuning and reproducible research, powering high-performing models in NLP.

87. Open-Platypus

- **Publisher**: Boston University
- **Size**: 24926 instances
- **License**: Mixed
- **Link**: `https://huggingface.co/datasets/garage-bAInd/Open-Platypus`
- **Description**: Open-Platypus is a composite English dataset containing 24,926 instruction-input-output examples across logic and reasoning tasks. Sourced from ten benchmarks—including PRM800K, MATH, ScienceQA, SciBench, ReClor, TheoremQA and Leetcode solutions—it employs sentence-transformer filtering to ensure <80% question similarity and removes 200 contaminated items. It supports refinement of large language models' logical reasoning and scientific problem-solving, serving as the core training corpus for Platypus2. License terms vary across components; see individual sources for details.

88. OpenPrompt

- **Publisher**: Tim Qian
- **Size**: 50 instances
- **License**: GPL-3.0
- **Link**: `https://github.com/timqian/openprompt.co`
- **Description**: OpenPrompt is a dynamic collection of the most popular prompts curated from OpenPrompt.co, updated daily to reflect trending and effective prompt engineering techniques. The dataset, available in JSON format, captures user preferences and evolving best practices for prompt design across diverse NLP applications.

89. Phoenix-sft-data-v1

- **Publisher**: The Chinese University of Hong Kong et al.
- **Size**: 464510 instances
- **License**: CC-BY-4.0
- **Link**: `https://huggingface.co/datasets/FreedomIntelligence/phoenix-sft-data-v1`
- **Description**: Phoenix-sft-data-v1 is a multilingual supervised fine-tuning dataset containing 464,510 samples, combining instruction-following and ChatGPT-distilled conversation data. It includes Alpaca-derived tasks, post-translated multilingual instructions, and user-centered prompts in 40 languages. The dataset also integrates ShareGPT and Discord-sourced dialogues. With nearly 1 million conversation turns and detailed multilingual annotations, it

supports multilingual language modeling, alignment, and chat adaptation. English and Chinese dominate the corpus, with broader linguistic diversity represented across the remaining data, enabling robust multilingual model training and evaluation.

90. PHYBench

- **Publisher**: Peking University et al.
- **Size**: 500 instances
- **License**: MIT
- **Link**: `https://huggingface.co/datasets/Eureka-Lab/PHYBench`
- **Description**: PHYBench is a 500-problems physics benchmark evaluating large language models' physical perception and multi-step reasoning across mechanics, electromagnetism, thermodynamics, optics, modern, and advanced physics. It offers 100 fully-annotated examples with handwritten solutions and 400 question-only items. Problems require symbolic, LaTeX-formatted answers assessed via the novel Expression Edit Distance (EED) metric for partial correctness. A rigorous three-stage validation pipeline ensures originality and clarity. PHYBench reveals substantial gaps between state-of-the-art models and human baselines and supports in-depth error analysis and leaderboard tracking.

91. PLM-Video Human

- **Publisher**: Meta FAIR et al.
- **Size**: 2797177 instances
- **License**: CC-BY-4.0
- **Link**: `https://huggingface.co/datasets/facebook/PLM-Video-Human`
- **Description**: PLM-Video Human is a large-scale human-annotated video understanding dataset for Vision-Language Model training, covering four tasks: fine-grained video question answering (FGQA) with 2.3M QA pairs, region-based video captioning (RCap), dense captioning (RDCap), and temporal localization (RTLoc). Each config provides annotated clip segments with questions, answers, captions, masks, start/end frames, and metadata drawn from diverse open-access sources. Released under CC-BY-4.0, PLM-Video Human supports detailed temporal, spatial, and semantic modeling of complex human activities across diverse realistic dynamic video scenarios.

92. PolyMath

- **Publisher**: Qwen Team et al.
- **Size**: 9000 instances
- **License**: Apache-2.0
- **Link**: `https://huggingface.co/datasets/Qwen/PolyMath`
- **Description**: PolyMath is a multilingual mathematical reasoning benchmark offering parallel problem sets in 18 languages across four difficulty tiers—K-12 to advanced mathematics—with splits labeled top, high, medium, and low. Each language contains 125 challenges per level, categorized by thought depth and knowledge breadth. The dataset ensures coverage of problem complexity and wide language representation, spanning over 75% of native speakers. High-quality translations validated by language experts guarantee clarity. PolyMath evaluates large language models' reasoning capabilities in diverse linguistic contexts.

93. PRISM

- **Publisher**: University of Oxford et al.
- **Size**: 77882 instances
- **License**: CC
- **Link**: `https://huggingface.co/datasets/HannahRoseKirk/prism-alignment`
- **Description**: The PRISM Alignment Dataset is a large-scale human feedback resource designed to assess preference and value alignment in large language models (LLMs). It consists of detailed survey responses from 1,500 participants across 75 countries, followed by multi-turn conversations with 21 LLMs. Participants rate model outputs on a 1-100 scale and provide fine-grained feedback, yielding 8,011 conversation trees and 68,371 scored utterances. The dataset includes four JSONL configurations—survey, conversations, utterances, and metadata—licensed under CC-BY and CC-BY-NC for research and educational use.

94. Prompt Engineering and Responses Dataset

- **Publisher**: Antrixsh Gupta
- **Size**: 5010 instances
- **License**: -
- **Link**: `https://www.kaggle.com/datasets/antrixsh/prompt-engineering-and-responses-dataset`
- **Description**: This dataset facilitates the study of prompt engineering by examining how different prompt types—questions, commands, and open-ended statements—influence generated text responses. With over 5,000 records, it enables analysis of prompt effectiveness across natural language generation, conversational agents, and sentiment influence.

95. Prompt Genius

- **Publisher**: Yan Lin, Haomin Wen, Zekai Shen
- **Size**: 2402 instances
- **License**: GPL-3.0
- **Link**: `https://www.promptgenius.site/`
- **Description**: PromptGenius is a comprehensive, multilingual prompt dataset structured by usage scenarios, facilitating efficient retrieval across domains like academic research, content creation, and office tasks. It continuously collects popular, high-quality prompts to enhance productivity and offers model output examples to improve prompt design.

96. Prompt Hackers

- **Publisher**: Prompt Hackers
- **Size**: 228 instances
- **License**: -
- **Link**: `http://www.prompthackers.co`
- **Description**: Prompt Hackers is an open platform for sharing prompts categorized across diverse domains including writing, music, marketing, health, gaming, education, coding, and business.

97. Prompt-in-context-learning

- **Publisher**: EgoAlpha Lab
- **Size**: 103 instances
- **License**: MIT
- **Link**: `https://github.com/EgoAlpha/prompt-in-context-learning`
- **Description**: Prompt-in-context-learning from EgoAlpha Lab offers an open-source engineering guide focused on mastering prompt engineering and in-context learning with large language models like ChatGPT, GPT-3, and FlanT5. Featuring a curated collection of 103 diverse prompts, it provides valuable, up-to-date resources for understanding how contextual prompts influence model behavior and performance.

98. PromptSet

- **Publisher**: University of Wisconsin-Madison
- **Size**: 93142 instances
- **License**: -
- **Link**: `https://github.com/pisterlabs/promptset`
- **Description**: PromptSet is a novel dataset containing over 61,000 unique developer-written prompts integrated within open-source Python projects. It highlights the emerging practice of structured prompting as a core component of application logic alongside traditional code.

99. PromptSource

- **Publisher**: Brown University et al.
- **Size**: 660 datasets
- **License**: Apache-2.0
- **Link**: `https://github.com/bigscience-workshop/promptsource`

- **Description**: PromptSource is a comprehensive toolkit designed for creating, sharing, and using natural language prompts, facilitating zero-shot and few-shot learning research with large language models. It hosts the Public Pool of Prompts (P3), containing around 2,000 English prompts for over 170 datasets. By providing a simple templating language (Jinja) and API, PromptSource enables reproducible prompt engineering and systematic evaluation, supporting advances in multitask fine-tuning and zero-shot generalization across diverse NLP tasks.

100. PubMedQA
    - **Publisher**: University of Pittsburgh et al.
    - **Size**: 273518 instances
    - **License**: MIT
    - **Link**: `https://huggingface.co/datasets/qiaojin/PubMedQA`
    - **Description**: PubMedQA is a biomedical question answering (QA) dataset designed to evaluate systems on their ability to answer yes/no/maybe research questions using corresponding PubMed abstracts. The dataset focuses on factual reasoning within biomedical literature.

101. QuickRef.ME
    - **Publisher**: Fechin
    - **Size**: 140 instances
    - **License**: GPL-3.0
    - **Link**: `https://quickref.me/chatgpt.html`
    - **Description**: QuickRef.ME is a prompt-sharing platform that compiles a comprehensive ChatGPT cheatsheet, aggregating prompts and usage tips from global sources. It serves as a practical resource for researchers and practitioners to understand effective prompt formulation and optimize interactions with large language models.

102. RedGPT-Dataset-V1-CN
    - **Publisher**: DA-southampton
    - **Size**: 50K instances
    - **License**: Apache-2.0
    - **Link**: `https://github.com/DA-southampton/RedGPT`
    - **Description**: RedGPT Dataset (V1-CN) offers 50,000 automatically generated multi-turn Chinese dialogues grounded in high-quality factual references from diverse domains such as history, science, law, and culture. Designed to enhance GPT models' factual accuracy, the dataset enables fine-tuning on realistic, knowledge-rich conversational data without costly manual annotation. It supports research in improving language models' truthfulness, dialogue generation, and knowledge integration.

103. RepLiQA
    - **Publisher**: ServiceNow Research et al.
    - **Size**: 71820 instances
    - **License**: CC-BY-4.0
    - **Link**: `https://huggingface.co/datasets/ServiceNow/repliqa`
    - **Description**: RepLiQA is a specialized QA dataset of 71,820 human-created Context-Question-Answer triplets from fictitious, natural-looking documents across 17 topics (e.g., local news, folklore, cybersecurity). Designed to test LLMs' ability to leverage novel reference texts without relying on memorized facts, each document includes five questions with 20% unanswerable. Fields include document IDs, topics, extracted text, questions, answers and long answers. Released under CC-BY-4.0 in four splits, RepLiQA supports question answering, text classification, topic retrieval and selective QA benchmarking.

104. ReTool-SFT
    - **Publisher**: ByteDance Seed
    - **Size**: 2000 instances
    - **License**: Apache-2.0
    - **Link**: `https://huggingface.co/datasets/JoeYing/ReTool-SFT`

- **Description**: ReTool is a reinforcement learning framework designed to teach large language models (LLMs) how to strategically use external computational tools during reasoning. By integrating tool-usage into the RL training loop, ReTool outperforms traditional text-only RL methods in accuracy and efficiency. Experiments on AIME2024 and AIME2025 benchmarks show it converges faster and achieves better results.

105. SciInstruct

- **Publisher**: The Knowledge Engineering Group et al.
- **Size**: 91750 instances
- **License**: CC-BY-4.0
- **Link**: `https://huggingface.co/datasets/zd21/SciInstruct`
- **Description**: SciInstruct is a large-scale scientific instruction dataset comprising 254,051 verified instructions across physics, chemistry, mathematics, and formal proofs (Lean). It addresses scientific reasoning challenges by collecting diverse questions from textbooks and problem sets, then generating high-quality step-by-step solutions using a multi-stage self-reflective annotation process powered by GPT-4.

106. Self-Instruct

- **Publisher**: University of Washington et al.
- **Size**: 52445 instances
- **License**: Apache-2.0
- **Link**: `https://huggingface.co/datasets/yizhongw/self_instruct`
- **Description**: Self-Instruct is an open Apache-2.0-licensed dataset and framework designed to enhance language models' instruction-following capabilities. It comprises four configurations: a self-generated set of 82K prompt-completion pairs produced via OpenAI's davinci engine; 50K samples from Super Natural Instructions; 52K prompts drawn from the P3 public pool; and 252 expert-crafted human evaluation tasks with associated inputs and outputs. All data is in English and supports instruction-tuning by providing diverse natural-language prompts paired with corresponding model or human completions. The dataset facilitates instruction-tuning.

107. ShareGPT4Video

- **Publisher**: University of Science and Technology of China et al.
- **Size**: 40178 instances
- **License**: CC-BY-NC-4.0
- **Link**: `https://huggingface.co/datasets/ShareGPT4Video/ShareGPT4Video`
- **Description**: ShareGPT4Video Captions Dataset offers a comprehensive collection of 4.8 million multimodal video captions generated by GPT4-Vision to improve alignment and fine-grained visual concept understanding in large video-language and text-to-video models. It comprises diverse subsets including the original 40K GPT4-Vision captions, 4,814K ShareCaptioner-Video outputs, and curated VQA and detailed caption mixes for supervised fine-tuning. Released under CC-BY-NC-4.0 in April 2024, it supports research in AIGC, computer vision, NLP, and multimodal AI development, bridging capabilities toward GPT4V and Sora benchmarks open-source releases.

108. ShareGPT90K

- **Publisher**: RyokoAI
- **Size**: 90K instances
- **License**: CC0
- **Link**: `https://huggingface.co/datasets/liyucheng/ShareGPT90K`
- **Description**: ShareGPT90K is a dataset of 90,665 conversational threads scraped from the ShareGPT platform. Each example includes a unique id and a sequence of messages, with each message annotated by its origin and its content.

109. ShareGPT-Chinese-English-90k

- **Publisher**: shareAI
- **Size**: 90K instances

- **License**: Apache-2.0
- **Link**: https://huggingface.co/datasets/shareAI/ShareGPT-Chinese-English-90k
- **Description**: ShareGPT-Chinese-English-90k is a 90K-instance bilingual parallel human-machine QA dataset covering real and complex user inquiries in both Chinese and English. Licensed under Apache-2.0, it provides semantically aligned Chinese-English QA pairs for robust training of instruction-following dialogue and text-generation models. Unlike synthetic API-simulated corpora, all questions originate from genuine user interactions, preserving realistic instruction distributions. Collected through voluntary sharing, it naturally filters out low-quality exchanges. The dataset supports question-answering and text-generation tasks and can be easily loaded via the Firefly framework.

110. Skywork-OR1-RL-Data

- **Publisher**: Skywork
- **Size**: 119112 instances
- **License**: -
- **Link**: https://huggingface.co/datasets/Skywork/Skywork-OR1-RL-Data
- **Description**: Skywork-OR1-RL-Data is a large-scale reinforcement learning dataset featuring 105,055 math problems and 14,057 coding questions curated for the Skywork-OR1 model series. Each example includes source attribution, structured prompts with roles, model-aware difficulty ratings for DeepSeek-R1 variants, and a reward model with ground truth and style labels. Problems are rigorously cleaned, deduplicated, and filtered by difficulty per variant. The dataset supports math and code splits totaling 1.5 billion bytes and facilitates robust reasoning training with rule-based RL recipes via curated pipelines efficiently.

111. Smart ChatGPT Prompts

- **Publisher**: Ashish Jaiswal
- **Size**: 26 instances
- **License**: MIT
- **Link**: https://github.com/asheeshcric/smart-chatgpt-prompts
- **Description**: Smart ChatGPT Prompts Awesome is a curated repository designed to enhance conversational AI development through carefully selected, effective prompts across diverse domains such as coding, academic writing, learning, and business.

112. SocialMaze

- **Publisher**: Xu Zixiang et al.
- **Size**: 200K instances
- **License**: CC-BY-4.0
- **Link**: https://huggingface.co/datasets/xzx34/SocialMaze
- **Description**: SocialMaze is a question-answering benchmark designed to evaluate large language models' social reasoning via hidden role deduction games. Each scenario presents a multi-agent setup where agents (Investigators, Criminal, Rumormongers, Lunatics) make public statements over three rounds. Models receive system prompts and dialogues, then must identify the true Criminal and Player 1's actual role. The dataset includes precise QA pairs, chain-of-thought reasoning, and supports easy (6-player) and hard (10-player) splits, facilitating fine-tuning, evaluation, and analysis of complex inference under deception. CC-BY-4.0 licensed.

113. SPIRIT

- **Publisher**: Dakuan Lu
- **Size**: 21639 instances
- **License**: MIT
- **Link**: https://huggingface.co/datasets/EricLu/System-Prompt-Instruction-Real-world-Implementation-Training-set

- **Description**: SPIRIT is a high-quality system prompt instruction dataset improving large language models' adherence to complex system prompts. It contains over 24,000 examples, including 3,000 real-world system prompts extracted from open-source GitHub repositories and 21,639 synthetically generated conversation samples via a multi-agent GPT-4-based pipeline. Following the OpenAI message format, SPIRIT ensures compatibility with fine-tuning workflows. Human evaluations show models fine-tuned on SPIRIT outperform instruct baselines in prompt compliance. Released under the MIT License, SPIRIT is ideal for enhancing system prompt following.

114. **SUPER-NATURAL INSTRUCTIONS**

- **Publisher**: Univ. of Washington et al.
- **Size**: 1616 datasets
- **License**: Apache-2.0
- **Link**: https://instructions.apps.allenai.org/
- **Description**: SUPER-NATURAL INSTRUCTIONS is a benchmark dataset designed to evaluate large language models' ability to generalize across diverse unseen tasks by leveraging natural language instructions. It emphasizes the importance of clear, comprehensive task descriptions to enable models to understand and perform novel tasks without additional training.

115. **The Cauldron**

- **Publisher**: Hugging Face et al.
- **Size**: 1880992 instances
- **License**: CC-BY-4.0
- **Link**: https://huggingface.co/datasets/HuggingFaceM4/the_cauldron
- **Description**: The Cauldron is a large-scale benchmark that aggregates the training splits of 50 public vision-language datasets. It covers diverse tasks such as general and text-based VQA, chart and figure understanding, table question answering, document OCR, captioning, visual reasoning, screenshot-to-code, and image-pair comparison. Each example comprises one or more images paired with user-assistant dialogues in a conversational Q&A format. Developed for fine-tuning the Idefics2 model, The Cauldron enables unified pretraining of architectures on a broad range of vision-language challenges and applications.

116. **The Prompt Index Prompt Database**

- **Publisher**: The Prompt Index
- **Size**: 620 instances
- **License**: -
- **Link**: https://thepromptindex.com/
- **Description**: The Prompt Index Prompt Database is a user-contributed repository featuring over 500 high-quality prompts spanning multiple domains, including SEO, content writing, coding, and more. This diverse dataset supports research in prompt engineering, cross-domain generalization, and AI-driven content generation.

117. **Tulu 3 SFT Mixture**

- **Publisher**: Allen Institute for AI et al.
- **Size**: 939344 instances
- **License**: ODC-BY-1.0
- **Link**: https://huggingface.co/datasets/allenai/tulu-3-sft-mixture
- **Description**: The Tulu 3 SFT Mixture is a 939k-example multilingual instruction-tuning corpus curated under the ODC-BY-1.0 license. It aggregates diverse supervised fine-tuning data—from crowdsourced, expert, and machine-generated sources—across over 70 languages. Composed of paired user-assistant dialogues with unique IDs and provenance labels, it blends samples from benchmarks like FLAN v2, CoCoNot, OpenAssistant, NuminaMath, WildChat, Table-GPT, and multiple Tulu 3 subsets. The single training split holds 939,343 examples. Designed to train Tulu-3 Llama-3.1 models through SFT, DPO, and RLHF.

118. UltraChat

- **Publisher**: Tsinghua University
- **Size**: 1468352 instances
- **License**: CC-BY-NC-4.0
- **Link**: `https://huggingface.co/datasets/stingning/ultrachat`
- **Description**: UltraChat is an open-source, large-scale multi-round conversational dataset generated using two ChatGPT Turbo APIs under an MIT license. It comprises 1-10 million English dialogue turns across three sectors: world knowledge queries, creative writing and content generation, and assistance on existing materials such as rewriting, summarization, and inference. By simulating user and assistant interactions with carefully designed prompts, UltraChat ensures diverse, high-quality exchanges. Generated conversations undergo rigorous post-processing and filtering to safeguard privacy and maintain robust, realistic dialogue for text-generation research.

119. UltraFeedback

- **Publisher**: Tsinghua University et al.
- **Size**: 63967 instances
- **License**: MIT
- **Link**: `https://huggingface.co/datasets/openbmb/UltraFeedback`
- **Description**: UltraFeedback is an MIT-licensed, open-source, large-scale preference dataset designed for training reward and critic models. It contains 64 K prompts drawn from UltraChat, ShareGPT, Evol-Instruct, TruthfulQA, FalseQA and FLAN, each answered by four out of 17 diverse LLMs under five alignment principles. The result is 256 K responses and 380 K fine-grained annotations covering instruction-following, truthfulness, honesty and helpfulness, all rated by GPT-4. Its scale, diversity and dense numerical plus textual feedback make it ideal for RLHF research and robust reward-model development.

120. UltraMedical

- **Publisher**: Tsinghua University
- **Size**: 409593 instances
- **License**: MIT
- **Link**: `https://huggingface.co/datasets/TsinghuaC3I/UltraMedical`
- **Description**: UltraMedical is a large-scale English biomedical instruction dataset featuring over 409,000 examples licensed under MIT. Each sample includes an identifier, instruction type, multi-turn conversation pairs between human queries and GPT-generated responses, a ground-truth answer, and a model-evaluated score. The training split comprises roughly 1.2 GB across 410K examples, sourced from both curated public data and synthetic augmentations. UltraMedical aims to support the development of specialized generalist models in biomedicine by providing diverse, high-quality instruction-response instances, and comprehensive evaluation metrics accompany each instance.

121. Universal Transformers Dataset

- **Publisher**: GoX AI
- **Size**: 1e24 datapoints
- **License**: -
- **Link**: `https://huggingface.co/datasets/future-technologies/Universal-Transformers-Dataset`
- **Description**: The Universal Transformer Dataset is a massive, scalable, multimodal resource comprising over one septillion structured datapoints across text, image, video and audio. Designed by the GoX AI Platform, it supports more than 40 NLP, vision, speech, and reinforcement learning tasks, covering over 200 languages. Preprocessed and pre-tokenized for efficient training, it is optimized for LLMs, vision, speech and multimodal architectures. Carefully curated and augmented via advanced AI models, it accelerates pretraining, fine-tuning, and zero-shot learning for cutting-edge AI research.

122. Unnatural Instructions

- **Publisher**: Tel Aviv University et al.

- **Size**: 240670 instances
- **License**: MIT
- **Link**: `https://huggingface.co/datasets/mrm8488/unnatural-instructions-full`
- **Description**: Unnatural Instructions is a large-scale dataset of automatically generated instruction-input-output triplets designed to facilitate instruction tuning of language models with minimal human effort. It contains over 240,000 examples, including original instructions, associated inputs, outputs, and optional constraints. Each instance also features multiple reformulations—paraphrased variants of instructions complete with inputs and outputs—to enhance model robustness. The publicly available training split comprises around 66,000 examples. This dataset supports research in instruction following, prompt paraphrasing, and evaluating model generalization across diverse complex tasks.

123. WebGLM-QA

- **Publisher**: Tsinghua University et al.
- **Size**: 44979 instances
- **License**: Apache-2.0
- **Link**: `https://huggingface.co/datasets/THUDM/webglm-qa`
- **Description**: WebGLM-QA is an English monolingual dataset designed for question answering and text generation, used to train the WebGLM generator. It contains 43,579 training samples, 1,000 validation examples, and 400 test instances. Each record pairs a user-posed question with a generated answer and a list of reference snippets that support the response. Hosted on Hugging Face, it provides a consistent structure—question, answer, references—enabling work on dialogue systems, retrieval-augmented generation, and answer justification.

124. Wizard_evol_instruct_196K

- **Publisher**: Microsoft et al.
- **Size**: 196K instances
- **License**: MIT
- **Link**: `https://huggingface.co/datasets/WizardLMTeam/WizardLM_evol_instruct_V2_196k`
- **Description**: Wizard_evol_instruct_196K is a MIT-licensed instruction-tuning dataset comprising 143K evolved QA pairs derived from Alpaca and ShareGPT. It represents an optimized version of the Evol-Instruct data used to train the WizardLM family of models. To assemble the complete instruction set of roughly 196K samples, users must merge this release with the original unfiltered ShareGPT dataset. The refined examples cover diverse conversational and instructional scenarios, facilitating improved alignment and performance in downstream open-source large language models, including structured prompts and responses.

125. Wizard_evol_instruct_70K

- **Publisher**: Microsoft et al.
- **Size**: 70K instances
- **License**: MIT
- **Link**: `https://huggingface.co/datasets/WizardLMTeam/WizardLM_evol_instruct_70k`

126. wonderful-prompts

- **Publisher**: LangGPT.ai
- **Size**: 108 instances
- **License**: MIT
- **Link**: `https://github.com/langgptai/wonderful-prompts`
- **Description**: wonderful-prompts is a curated collection of high-quality Chinese ChatGPT prompts designed to enhance usability and creativity in conversational AI applications. It offers diverse prompt templates covering coding, writing, productivity, art, and specialized expert roles, supporting research on prompt engineering and natural language interaction.

127. xP3

- **Publisher**: Hugging Face et al.
- **Size**: 82 datasets
- **License**: Apache-2.0
- **Link**: `https://huggingface.co/datasets/bigscience/xP3`
- **Description**: xP3 (Crosslingual Public Pool of Prompts) is a multilingual prompt and dataset collection spanning 46 languages and 13+ NLP tasks (e.g., QA, translation, summarization, code generation). Assembled from expert-generated and crowdsourced annotations under an Apache-2.0 license, it supports zero-shot and instruction-tuning for models like BLOOMZ and mT0. The training mixture covers closed-book and extractive QA, multiple-choice, paraphrase identification, program synthesis, sentiment analysis, structure-to-text, summarization, classification and more, totaling over 788 million samples. xP3 streamlines reproducible multilingual finetuning across diverse data scales.

128. ZeroSearch_dataset
- **Publisher**: Tongyi Lab et al.
- **Size**: 172740 instances
- **License**: Apache-2.0
- **Link**: `https://huggingface.co/datasets/sunhaonlp/ZeroSearch_dataset`
- **Description**: The ZeroSearch_dataset is a benchmark designed to evaluate and enhance large language models' search capabilities without performing external retrieval. Released under the Apache-2.0 license, it targets question-answering tasks that require models to infer answers using internal knowledge rather than querying outside sources. Created alongside the ZeroSearch framework, the dataset fuels research on incentivizing retrieval-like reasoning within LLMs. Researchers can obtain the dataset and related materials from the project page to benchmark model performance and spur advances in robust knowledge retrieval.

129. Zhihu-KOL
- **Publisher**: wangrui6
- **Size**: 1006218 instances
- **License**: MIT
- **Link**: `https://huggingface.co/datasets/wangrui6/Zhihu-KOL`
- **Description**: Zhihu-KOL is a large-scale Chinese question-answering dataset derived from the Zhihu platform, designed for training open-domain assistants. It comprises 1,006,218 training instances of instruction-response pairs, each annotated with source and metadata fields.

## F  ADDITIONAL EXPERIMENTAL RESULTS

### F.1  TOKEN-LEVEL ANALYSIS

In this section, we provide the comparision of 3/4/5-grams for all datasets (except ShareGPT, which is displayed in the main paper) in Figure 6, and the top-5 $n$-grams comparison across datasets in Figure 7.

The conclusions drawn from these figures are consistent with the main paper, for example, high-frequency $n$-grams phrases mainly show command sentences and topic content. In addition, there are two other findings:

1. The $n$-grams phrases of some datasets include abnormal content (e.g. "*identify which instrument be string*" in dolly-15 and "*The quick brown fox jumps over the lazy dog*" in Self-Instruct), which indicates that there is a lot of repetition in the input content of the template tasks or some instructions used to construct the dataset, which may affect the balance of the dataset.

2. Some $n$-grams phrases extracted from fixed sentences show convolution-like effects, such as "*The quick brown fox jumps over the lazy dog*" is segmented into 5-grams phrases such as "*quick brown fox jump over*", "*jump over the lazy dog*", etc.

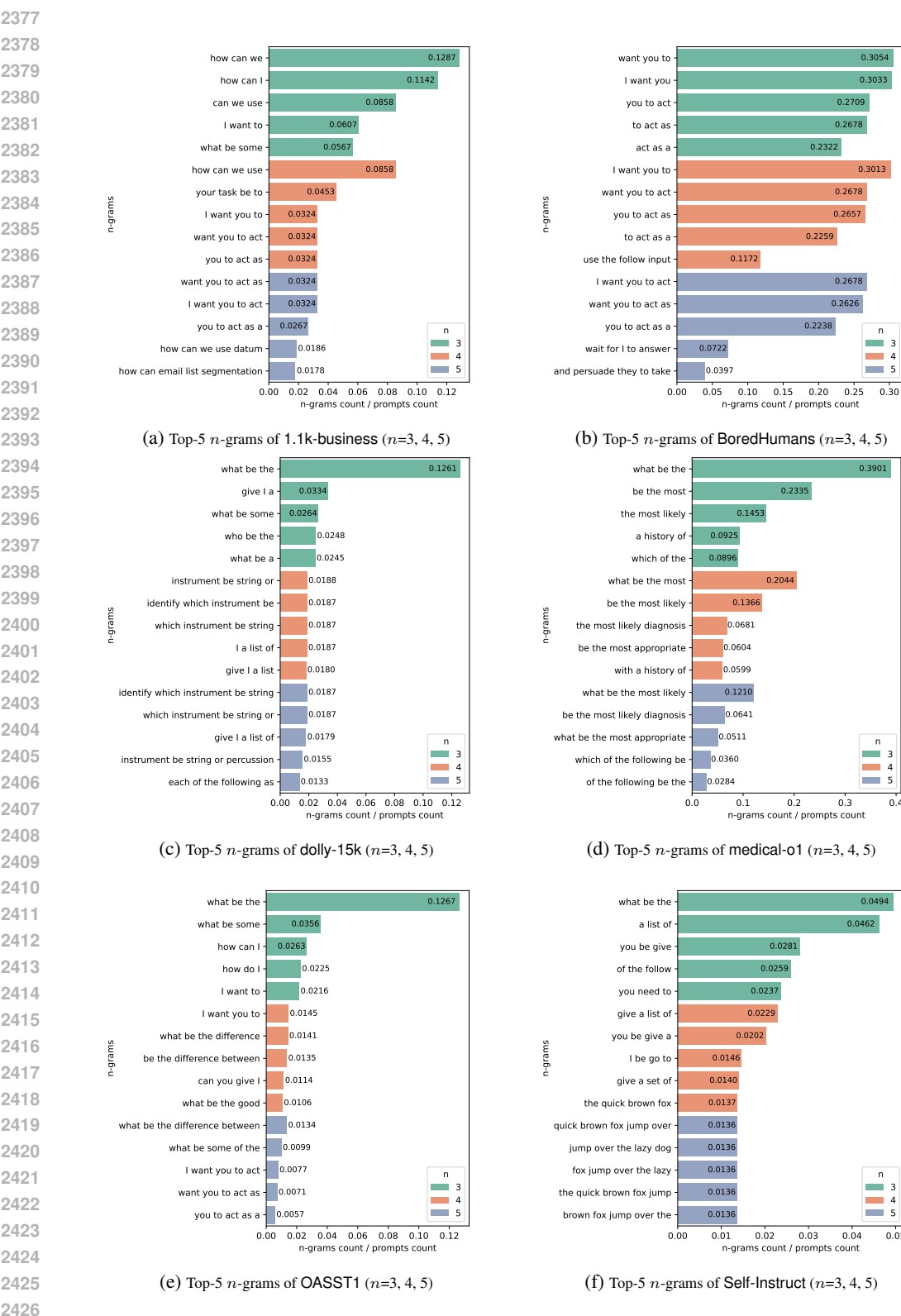

Figure 6: Comparison of 3/4/5-grams in the same dataset

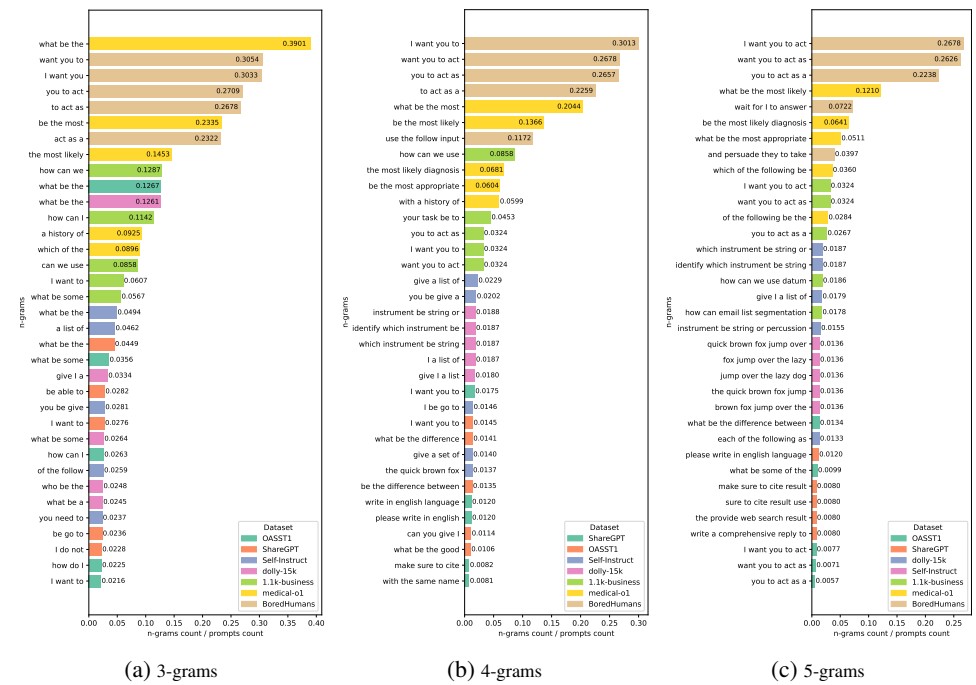

(a) 3-grams      (b) 4-grams      (c) 5-grams

Figure 7: Top-5 $n$-grams comparison across datasets

## F.2 SYNTACTIC-LEVEL ANALYSIS

In this section, we present the complete experimental data for all identified dependency types, along with their proportions in the datasets, as shown in Table 5. Additionally, Table 6 lists all detected Part-of-Speech tags and their corresponding proportions. Figure 8 further illustrates the ten most common verbs and their top five direct noun objects found in the prompt datasets except medical-o1 and ShareGPT, which are shown in the main paper.

These additional data further support our conclusions. **(1)** The medical-o1 dataset, which consists of professionally crafted medical prompts, exhibits a relatively high proportion of numerical modifiers (nummod, 0.0276) and passive auxiliaries (auxpass, 0.0101) in dependency analysis, as well as a notably high usage of numerals (NUM, 0.0309) in POS tagging. These features reflect a terminology-dense and precision-oriented language style that emphasizes processes and outcomes rather than agents. **(2)** In the 1.1k-business dataset, the verb-noun pairs reflect language commonly used in business contexts, such as *"create plan"* and *"create strategy"*. In contrast, the verb-noun pairs observed in BoredHumans, OASST1, and Self-Instruct suggest more generic and broadly applicable usage scenarios.

Anomalously, in the dolly-15k dataset, the most frequent verb-noun pairs exhibit a skewed distribution, with the highest-frequency nouns overwhelmingly associated with only the top one or two verbs. Moreover, these frequent verb-noun pairs often lack clear task-specific semantics—for example, *"tell i"*, *"give list"*, and *"classify each"*. This pattern may be attributed to the manual generation process, which is susceptible to the individual linguistic habits of annotators.

## F.3 SEMANTIC-LEVEL ANALYSIS

In this section, we show the distribution of sampled embedding points after PCA for all datasets (except for medical-o1 and Self-Instruct, which are shown in the main paper) in Figure 9.

We can still observe from the results that datasets with more concentrated topical focus (e.g., 1.1k-business) exhibit clear clustering patterns, whereas those with broader thematic coverage (e.g., ShareGPT) display a more dispersed distribution of data points.

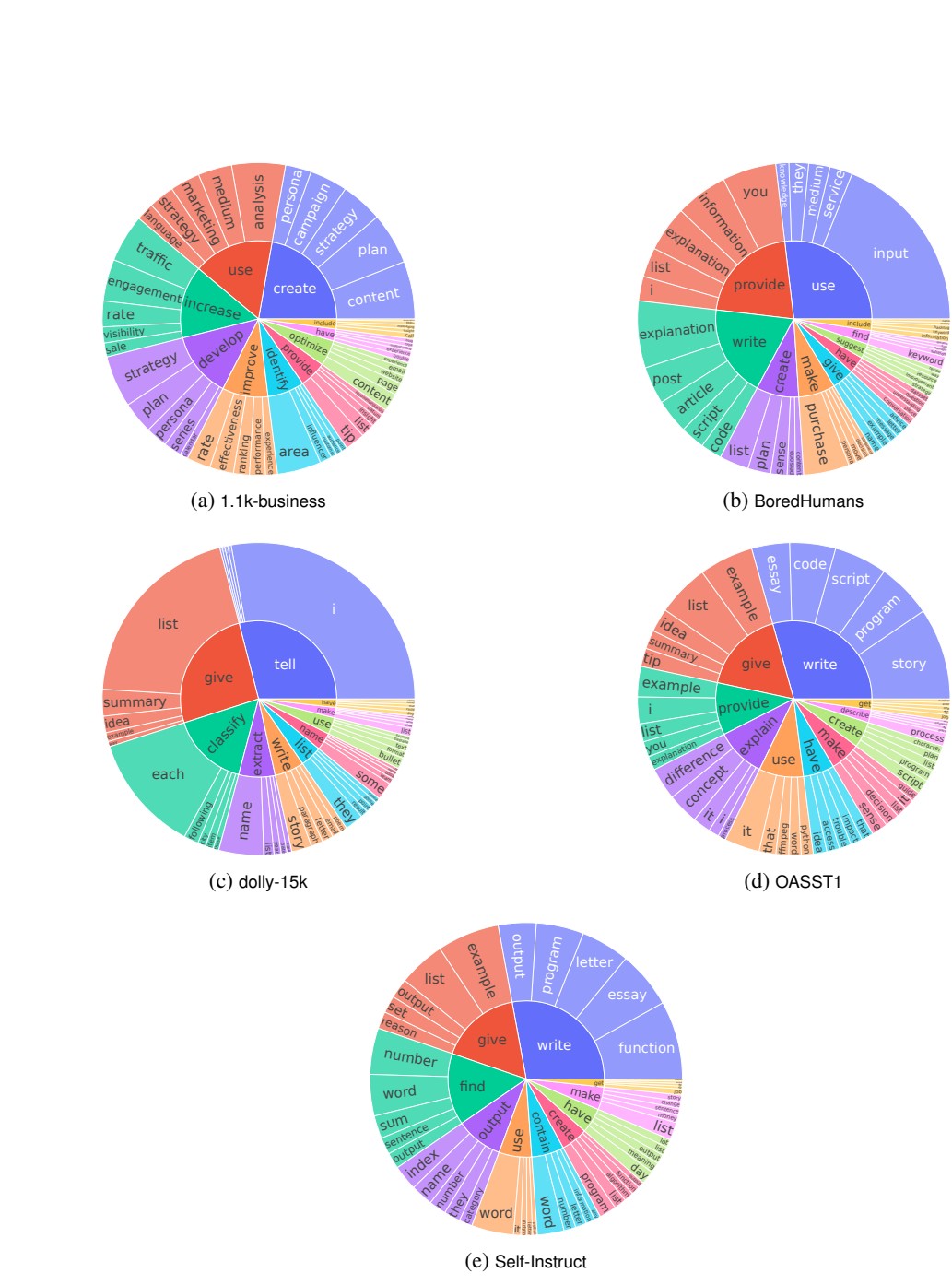

Figure 8: The top-10 most common verbs and their top-5 direct noun objects in prompt datasets.

Table 5: All detected dependency types, with the values indicating their proportions in the dataset. '-' means the Dependency Type not detected in the dataset.

| Dependency Type | 1.1k-business | BoredHumans | dolly-15k | medical-o1 | OASST1 | Self-Instruct | ShareGPT |
|---|---|---|---|---|---|---|---|
| punct | 0.1227 | **0.1985** | 0.1445 | 0.1216 | 0.1273 | 0.1863 | 0.154 |
| prep | 0.0759 | 0.0672 | 0.0866 | **0.1013** | 0.0816 | 0.0676 | 0.0764 |
| det | 0.0518 | 0.0692 | **0.0961** | 0.0906 | 0.0841 | 0.0838 | 0.0693 |
| pobj | 0.0718 | 0.062 | 0.0817 | **0.0979** | 0.076 | 0.0645 | 0.0711 |
| nsubj | 0.0596 | 0.0545 | 0.065 | 0.0469 | **0.0739** | 0.0596 | 0.0562 |
| ROOT | 0.0528 | 0.0462 | 0.0768 | 0.0444 | 0.0604 | **0.0792** | 0.0437 |
| amod | 0.0573 | 0.0527 | 0.0469 | **0.1072** | 0.0523 | 0.0384 | 0.048 |
| dobj | **0.0904** | 0.0665 | 0.0447 | 0.0315 | 0.0594 | 0.057 | 0.0519 |
| compound | **0.0742** | 0.0471 | 0.0719 | 0.0716 | 0.0436 | 0.023 | 0.0576 |
| conj | 0.0457 | 0.0494 | **0.0569** | 0.0391 | 0.0343 | 0.0359 | 0.0371 |
| aux | **0.0642** | 0.0425 | 0.0257 | 0.0143 | 0.0495 | 0.0302 | 0.0355 |
| dep | 0.0095 | 0.0306 | 0.007 | 0.0183 | 0.0218 | **0.0611** | 0.0577 |
| cc | **0.04** | 0.0297 | 0.0203 | 0.0291 | 0.0289 | 0.0203 | 0.0287 |
| advmod | 0.0269 | 0.0273 | 0.0263 | 0.023 | **0.0383** | 0.0224 | 0.0299 |
| poss | **0.0401** | 0.0183 | 0.0084 | 0.0132 | 0.0118 | 0.0124 | 0.0116 |
| appos | 0.003 | 0.0223 | 0.017 | 0.0099 | 0.0129 | 0.0207 | **0.0238** |
| attr | 0.0044 | 0.005 | **0.0306** | 0.014 | 0.0163 | 0.0113 | 0.0083 |
| nummod | 0.003 | 0.0073 | 0.0096 | **0.0276** | 0.0093 | 0.0136 | 0.0155 |
| nmod | **0.0252** | 0.0126 | 0.0042 | 0.0095 | 0.0068 | 0.0043 | 0.0126 |
| ccomp | 0.0058 | 0.0129 | 0.0089 | 0.0044 | 0.013 | **0.0142** | 0.013 |
| relcl | **0.0146** | 0.0088 | 0.0097 | 0.0072 | 0.0109 | 0.0097 | 0.0094 |
| xcomp | **0.0194** | 0.0104 | 0.0042 | 0.0042 | 0.0106 | 0.0079 | 0.0099 |
| advcl | 0.0104 | 0.0109 | 0.0056 | 0.0066 | 0.0115 | 0.0086 | **0.0122** |
| npadvmod | 0.0026 | 0.0068 | 0.0052 | **0.0141** | 0.0051 | 0.0052 | 0.0066 |
| acomp | 0.0034 | 0.0041 | 0.0059 | 0.007 | 0.0086 | **0.0091** | 0.0068 |
| mark | 0.002 | 0.0057 | 0.0038 | 0.0033 | 0.0087 | **0.0114** | 0.0095 |
| acl | 0.0038 | 0.0062 | 0.0055 | 0.0077 | 0.0056 | **0.0079** | 0.0063 |
| auxpass | 0.0016 | 0.0017 | 0.0062 | **0.0101** | 0.0055 | 0.0052 | 0.0059 |
| pcomp | **0.008** | 0.0048 | 0.0036 | 0.0053 | 0.0057 | 0.0035 | 0.0051 |
| nsubjpass | 0.0014 | 0.0015 | 0.005 | **0.009** | 0.0046 | 0.0048 | 0.0051 |
| neg | 0.0015 | 0.0031 | 0.0016 | 0.002 | 0.0043 | 0.0033 | **0.0045** |
| case | **0.0035** | 0.0013 | 0.0032 | 0.0024 | 0.002 | 0.0011 | 0.0023 |
| dative | 0.0003 | 0.0029 | **0.0041** | 0.0001 | 0.0035 | 0.0023 | 0.0016 |
| prt | 0.0014 | 0.0018 | 0.0014 | 0.0006 | 0.0025 | **0.0047** | 0.0025 |
| intj | 0.0004 | **0.0038** | 0.0012 | 0.0002 | 0.0033 | 0.0016 | 0.0025 |
| agent | 0.0002 | 0.0003 | 0.001 | **0.002** | 0.001 | 0.0015 | 0.0012 |
| expl | 0.0 | 0.0002 | 0.0004 | 0.0007 | **0.0016** | 0.0014 | 0.0011 |
| quantmod | 0.0 | 0.0002 | 0.0005 | 0.0007 | 0.0009 | 0.0014 | **0.0016** |
| meta | 0.0001 | 0.0016 | 0.0001 | 0.0 | 0.0003 | **0.0018** | 0.0012 |
| oprd | 0.0002 | **0.0009** | **0.0009** | 0.0007 | **0.0009** | 0.0004 | 0.0008 |
| predet | - | 0.0003 | 0.0005 | 0.0001 | 0.0006 | **0.0009** | 0.0005 |
| csubj | 0.0005 | 0.0001 | 0.0004 | 0.0004 | 0.0005 | 0.0001 | **0.0007** |
| parataxis | - | **0.0009** | 0.0001 | 0.0 | 0.0003 | 0.0002 | 0.0006 |
| preconj | 0.0 | 0.0001 | **0.0008** | 0.0002 | 0.0002 | 0.0002 | 0.0003 |
| csubjpass | 0.0 | - | **0.0001** | 0.0 | 0.0 | 0.0 | 0.0 |

Table 6: All detected Parts-of-Speech Tags, with each value indicating its proportion in a dataset. '-' means the POS tag not detected in the dataset.

| POS | 1.1k-business | BoredHumans | dolly-15k | medical-o1 | OASST1 | Self-Instruct | ShareGPT |
|---|---|---|---|---|---|---|---|
| NOUN | **0.2637** | 0.2103 | 0.1899 | 0.259 | 0.1946 | 0.2027 | 0.1944 |
| PUNCT | 0.1094 | **0.1942** | 0.1435 | 0.1158 | 0.1231 | 0.1839 | 0.145 |
| VERB | **0.1302** | 0.1094 | 0.0871 | 0.0775 | 0.1069 | 0.0999 | 0.0979 |
| ADP | 0.0758 | 0.0678 | 0.0858 | **0.0998** | 0.0851 | 0.0701 | 0.0789 |
| DET | 0.0506 | 0.0693 | **0.0949** | 0.0893 | 0.0839 | 0.0844 | 0.0696 |
| PRON | **0.0912** | 0.0708 | 0.0695 | 0.0369 | 0.0869 | 0.0701 | 0.0583 |
| ADJ | 0.0588 | 0.0543 | 0.0538 | **0.1104** | 0.0632 | 0.0498 | 0.0563 |
| PROPN | 0.0219 | 0.0372 | **0.1272** | 0.0515 | 0.0471 | 0.0294 | 0.0703 |
| AUX | 0.0458 | 0.0379 | 0.0608 | 0.0382 | **0.0644** | 0.0453 | 0.0423 |
| CCONJ | **0.0399** | 0.0294 | 0.0209 | 0.0291 | 0.0288 | 0.0204 | 0.0286 |
| SPACE | - | 0.0267 | 0.0053 | 0.0175 | 0.019 | 0.0504 | **0.0517** |
| PART | **0.0358** | 0.0223 | 0.013 | 0.01 | 0.0222 | 0.0172 | 0.0213 |
| NUM | 0.0041 | 0.0146 | 0.014 | **0.0309** | 0.0153 | 0.0282 | 0.0273 |
| ADV | 0.0097 | **0.0259** | 0.0107 | 0.0209 | 0.0238 | 0.0153 | 0.0247 |
| SCONJ | 0.0199 | 0.0105 | 0.0198 | 0.007 | **0.0242** | 0.0197 | 0.0152 |
| X | **0.035** | 0.0128 | 0.0015 | 0.0005 | 0.0044 | 0.0075 | 0.0082 |
| SYM | **0.008** | 0.0037 | 0.0008 | 0.0049 | 0.0035 | 0.0031 | 0.0073 |
| INTJ | 0.0002 | 0.003 | 0.0012 | 0.001 | **0.0037** | 0.0027 | 0.0028 |

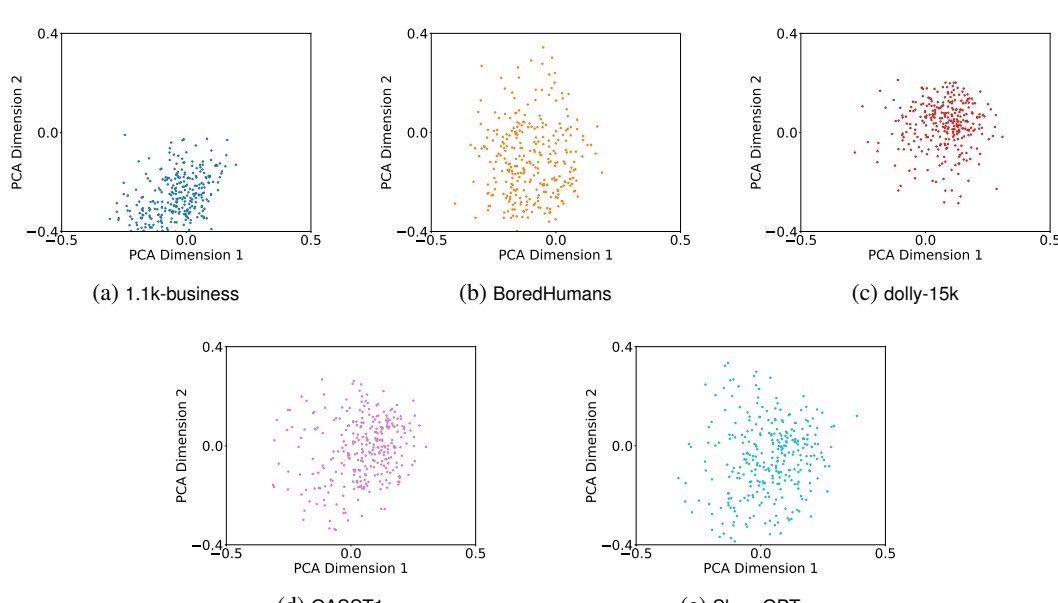

(a) 1.1k-business      (b) BoredHumans      (c) dolly-15k

(d) OASST1      (e) ShareGPT

Figure 9: Semantic prompt embeddings distribution for all other datasets.

## F.4 APPLICATION

In this section, we present detailed information for two case studies illustrating the effects of prompt optimization on language model responses. For each case, we include the prompts used both before and after optimization, as well as the responses generated by the language model for each version. The two questions are selected from PRM800K (Lightman et al., 2023) and MMLU-Pro (Wang et al., 2024c)

# Seating Problem

> Source: OpenAI/PRM800K

## Original Prompt

In how many ways can 8 people be seated around a square table with 2
   people on a side? (Two configurations are considered equivalent if
   one is a rotation of another.)

## Optimized Prompt

How many different ways can we arrange 8 people around a square table,
   with 2 people seated on each side? Remember, two arrangements are
   considered the same if one can be obtained from the other by rotating
    the table.

## Answer of Original Prompt

To solve the problem of seating 8 people around a square table with 2
   people on each side, where configurations that can be obtained by
   rotation are considered equivalent, we can follow these steps:

1. **Identify the arrangement of people around the table:**
  The square table has 4 sides and 2 seats on each side, resulting in a
     total of 8 seats. We need to count the distinct ways to arrange 8
     people in these 8 seats.

2. **Consider the rotational symmetry:**
  Since configurations that can be obtained by rotating the arrangement
     are considered the same, we need to divide the total arrangements
     by the number of unique rotations.

3. **Fix one person to eliminate equivalency due to rotations:**
  We can fix one person in one of the seats to simplify our calculations.
      Fixing one person creates a reference point and allows us to treat
      the remaining arrangements as linear (ignoring the rotational
     symmetry now).

4. **Arrange the remaining people:**
  With one person fixed, there are 7 remaining people who can fill the
     other 7 seats. The number of ways to arrange 7 people is given by
     \(7!\):
  \[
  7! = 5040
  \]

5. **Result:**
  Since fixing one person effectively removes any ambiguity due to
     rotations, we do not need to divide by the number of rotations (
     which was initially 4 for a square). Thus, the total number of
     distinct arrangements of 8 people around the table is simply \(7!\).

In conclusion, the number of ways to seat 8 people around a square table
   with 2 people on each side, considering rotations as equivalent, is:

$$
\boxed{5040}
$$

## Answer of Optimized Prompt

To find the number of different ways to arrange 8 people around a square table with 2 people seated on each side, we need to consider the following steps:

1. **Total Arrangements without Considering Rotations**: First, we calculate the total arrangements of the 8 people without any restrictions on rotations. For 8 people, the total number of arrangements is given by the factorial of the number of people:
$$
8! = 40320
$$

2. **Considering Rotations**: Since the table can be rotated, we need to account for these symmetries. The table can be rotated in 4 ways (0, 90, 180, and 270). Therefore, we need to divide the total arrangements by the number of rotations:
$$
\text{Distinct arrangements} = \frac{8!}{4}
$$
Substituting the value we calculated for $(8!)$:
$$
\text{Distinct arrangements} = \frac{40320}{4} = 10080
$$

Thus, there are $(10080)$ different ways to arrange 8 people around a square table with 2 people seated on each side where arrangements that can be obtained from each other through rotation are considered identical.

The final answer is:
$$
\boxed{10080}
$$

# Swimming Mechanism

> Source: TIGER-Lab/MMLU-Pro

## Original Prompt:

What is the major difference between themethodof swimming in a fish such as a tuna, and that of a dolphin?

Options:
{
"A": "Both use side to side tail fin movement for swimming",
"B": "Both species swim using rapid oscillation of their dorsal fins.",
"C": "Tuna use their gills for propulsion while dolphins use their blowhole to aid in swimming.",
"D": "Tuna swim by contracting their bodies and pushing water backwards, while dolphins swim primarily by using their pectoral fins.",
"E": "Dolphins have a flexible backbone that allows for up-and-down movement, while tuna do not have a backbone.",
"F": "Dolphins swim by rotating their entire body in a spiral motion, while tuna swim by flapping their fins only.",
"G": "Tuna propel themselves using jet propulsion by expelling water, whereas dolphins glide through the water without any propulsion.",
"H": "Dolphins use their hind limbs for swimming",

```
"I": "In tuna, the tail fin is oriented dorso-ventrally and movement is
    side to side, while in dolphins, the tail fin is oriented laterally
    with dorso-ventral movement. Dolphins also have vestigial hind limbs
    and modified forelimbs for balance, unlike tuna.",
"J": "Tuna have modified forelimbs for balance"
}
```

## Optimized Prompt

What is the key difference in how a tuna swims compared to a dolphin?

```
Options:
{
"A": "Both use side-to-side tail fin movements to swim.",
"B": "Both species utilize rapid oscillations of their dorsal fins for
    swimming.",
"C": "Tuna propel themselves using their gills, while dolphins rely on
    their blowhole to assist in swimming.",
"D": "Tuna swim by contracting their bodies and pushing water backward,
    whereas dolphins primarily use their pectoral fins for movement.",
"E": "Dolphins have a flexible backbone that allows for up-and-down
    movement, unlike tuna which lack this flexibility.",
"F": "Dolphins swim by rotating their entire bodies in a spiral motion,
    while tuna mainly flap their fins.",
"G": "Tuna use jet propulsion by expelling water, while dolphins glide
    smoothly through the water.",
"H": "Dolphins swim using their hind limbs.",
"I": "In tuna, the tail fin is oriented dorso-ventrally and moves side to
     side, while in dolphins, the tail fin is oriented laterally and
    moves up and down. Additionally, dolphins have vestigial hind limbs
    and modified forelimbs for balance, which tuna do not.",
"J": "Tuna have modified forelimbs for balance."
}
```

## Answer of Original Prompt

The major difference between the swimming methods of a fish such as a
    tuna and a dolphin can be best captured by option:

**D: "Tuna swim by contracting their bodies and pushing water backwards,
    while dolphins swim primarily by using their pectoral fins."**

This option accurately describes the fundamental difference in their
    swimming mechanics, as tuna are fish that primarily use the lateral
    movement of their tails (caudal fins) and body contractions to move
    through the water, while dolphins, being mammals, utilize their
    pectoral fins along with the up-and-down movement of their tail (
    fluke) to propel themselves.

## Answer of Optimized Prompt

The key difference in how a tuna swims compared to a dolphin is best
    expressed by option:

**I**: "In tuna, the tail fin is oriented dorso-ventrally and moves side
    to side, while in dolphins, the tail fin is oriented laterally and
    moves up and down. Additionally, dolphins have vestigial hind limbs
    and modified forelimbs for balance, which tuna do not."

This option correctly describes the anatomical differences in tail fin
    structure and swimming mechanics between the two species.

