# OpenReview forum: "Large Language Model Prompt Datasets: An In-depth Analysis and Insights"
_ICLR.cc/2026/Conference — ICLR 2026 Conference Withdrawn Submission_

### Official Review · Reviewer_n4md · 2025-10-27

**Soundness:** 3
**Presentation:** 4
**Contribution:** 3
**Rating:** 8
**Confidence:** 4

**Summary:**

The paper introduces a large-scale prompt dataset with over 600M prompts from many diverse data sources, along with hte corresponding code for extending the dataset in the future. They analyze features of the dataset, and hierarchically cluster across source, content, and target. Finally, they contribute a prompt engineering method based on information from their prompt dataset.

**Strengths:**

S1: The paper carries out a large-scale compilation and analysis of prompt datasets, encompassing over 1.2BT of data and over 674M prompts. The underlying datasets and prompts (along with the code to extract and clean them) could serve as a great resource for future work.
S2: The authors introduce a useful hierarchy for the prompt dataset, showing diversity and breadth.
S3: I found the seven representative datasets (section 5) and corresponding analysis quite useful for a case study

**Weaknesses:**

W1: I found Section 6 - Applciation with the prompt engineering technique to be insufficiently fleshed out. As far as I can tell, the only empirical evidence for this contributeion is a single example that was corrected after prompt optimization (Figure 5). In my opinion, unless it is overhauled with many more details and empirical backing, this section and contribution should be omitted from the paper.

Minor points / potential typos:
- Several of the citations are quite strange (eg 90-91, "(hug)", "(kag)"). I understand that the reason for this is likely that they are non-traditional citation sources (urls instead of papers). However, for some of these, there may be some underlying paper you could cite (e.g., https://arxiv.org/abs/2109.02846 for Huggingface Datasets). Could the authors go through this and try to add in paper citations where they exist?
- Figure 1 is fantastic. However, some minor possible improvements: 1) are there not three hierarchies in one chart here? It may be less confusing to present them as such, instead of a single hierarchy. 2) Might it be possible to show the percentage of the data that falls into each category, with eg a sankey diagram? Not at all necessary, but could add some legibility.

**Questions:**

Q1: How is "higher-performing prompts" defined in Section 6? Is the assumption just that the prompts in the dataset are higher performing than other hypothetical prompts?
Q2: Do you have any empirical evidence for the prompt optimizatino in section 6?

---

### Official Review · Reviewer_aP7G · 2025-10-30

**Soundness:** 2
**Presentation:** 2
**Contribution:** 2
**Rating:** 2
**Confidence:** 3

**Summary:**

This paper addresses the gap in the systematic analysis of Large Language Model (LLM) prompt datasets. The authors conduct a comprehensive data discovery process, compiling a catalog of 129 prompt datasets totaling over 673M instances from diverse sources. The paper not only introduces a hierarchical taxonomy for LLM prompt datasets, performing an in-depth linguistic analysis on seven representative examples to highlight structural differences from general text, but also tries to propose a novel prompt optimization method that guides prompt generation toward a syntactic "centroid".

**Strengths:**

1.The paper's most significant contribution is its systematic and large-scale data discovery process. The authors have aggregated a massive collection of 129 datasets from varied channels, including academic platforms, open-source repositories like GitHub, and prompt-sharing websites. The resulting catalog (Appendix E) is a valuable resource for the research community.

2.The proposed hierarchical taxonomy (Figure 1) is well-structured and thorough. It provides a much-needed standardized framework for classifying prompt datasets, covering critical dimensions like Generation Process (e.g., human- vs. model-generated), Prompt Engineering (e.g., Few-shot, CoT), and Downstream Tasks (e.g., model-oriented, data-oriented).

3.The paper moves beyond a simple enumeration of datasets by conducting a rigorous, multi-level linguistic analysis (Section 5). The comparison of syntactic features (e.g., dependency types, POS tags) across datasets like medical-01 and 1.1k-business provides novel insights into the unique linguistic properties of "prompt language" and how it diverges based on domain.

**Weaknesses:**

1.Weak Justification for the "Centroid" Optimization: The link between the linguistic analysis (Section 5) and the proposed application (Section 6) is tenuous. The analysis in Section 5 is descriptive, identifying features of different datasets. However, the optimization method in Section 6 is built on the assumption that a "centroid" of syntactic patterns is “associated with higher-performing prompts”. The paper provides no experimental evidence to validate this core hypothesis—it is not demonstrated that this syntactic centroid actually correlates with better model responses.

2.Lack of "Prompt Effect" Evaluation: As a paper analyzing datasets, its key missing piece is an evaluation of prompt effectiveness. The study remains purely descriptive, analyzing the form of prompts and presenting some statistical results, without connecting it to the quality of the output. A stronger paper would have (at minimum) sampled prompts based on the identified linguistic features (e.g., high amod frequency in medical-01 or high dobj in 1.1k-business) and measured their impact on task performance. This omission reduces the practical utility of the linguistic findings.

3.Vague and Irreproducible Optimization Method: The prompt optimization method described in Section 6 is too vague to be reproducible. The paper states it takes an "average of the high-dimensional embeddings of POS tags and dependency relations" to define a centroid, but does not specify the embedding model, dimensionality, or averaging technique. Furthermore, it claims an LLM is "guided to rewrite the prompt", but the (meta) prompt or guidance mechanism used for this rewriting is not provided, making the case study in Figure 5 anecdotal rather than scientific.

4.Misalignment with Main Conference Scope: The paper's core strength lies in its comprehensive data collection and detailed descriptive taxonomy (Figure 1), which are hallmarks of a strong "Datasets and Benchmarks" contribution. However, it struggles to present a compelling, hypothesis-driven research contribution suitable for a main conference track. The linguistic analysis (Section 5) is purely observational , and the novel "application" (Section 6) is presented as an underdeveloped sketch without rigorous empirical validation. As such, the work reads more like an excellent analytical report on what exists rather than a research paper proposing and validating a new method or finding that advances the state-of-the-art. The work might be better positioned for a specialized "Datasets and Benchmarks" track rather than the main technical track of a conference like ICLR.

**Questions:**

see weakness

---

### Official Review · Reviewer_Ez5X · 2025-10-31

**Soundness:** 2
**Presentation:** 2
**Contribution:** 2
**Rating:** 2
**Confidence:** 4

**Summary:**

This paper presents a large-scale survey and analysis of prompt datasets. The authors have compiled a collection of 129 datasets and propose a hierarchical taxonomy to categorize them based on three dimensions: Source, Content, and Target .
Following this, the paper conducts an in-depth linguistic analysis (lexical, syntactic, and semantic) on seven representative datasets. This analysis aims to uncover the linguistic characteristics of "prompts" as a unique text genre and contrast them with traditional corpora.
Finally, based on this analysis, the authors propose an "application" for prompt optimization, which involves defining a "centroid" by calculating the average syntactic structure of the analyzed prompt corpus and then guides an LLM to rewrite new prompts to align their syntactic structure with this centroid, purportedly to improve the quality of model outputs.

**Strengths:**

1. The primary and most solid contribution of this paper lies in Sections 3 & 4. The authors have invested significant effort in prompt dataset collection (129 heterogeneous sources) and taxonomy construction. The collected dataset and taxonomy provides a much-needed map and index for the field of prompt datasets, which is a valuable resource.

2. The analysis in Section 5 present an attempt to systematically analyze the linguistic features of prompts, from wording to syntax to sentence-level semantics. It empirically confirms several intuitions, such as prompts being "inquiry- or command-focused" and that domain-specific prompts (e.g., medical ) have unique linguistic "fingerprints".

**Weaknesses:**

1. Prompting is a very important field in todays' LLM application. In Prompt Engineering of real application, the core object of interest and optimization are often in complex, thousand-token system prompt/meta-prompt or long prompts for complex task. However, the objects of collection and analysis in this paper are only short, task-focused "queries" or "Instructions" (such as the self-instruct, Alpaca, GSM8K datasets, and most datasets presented in the appendix), which has a large gap from real applications.
    - This, first and foremost, significantly reduces the practical utility and real-world contribution of this paper.
    - For the complex prompts that the field actually focuses on, the syntactic analysis in Section 5 is not applicable. In fact, most of this analysis is shallow, can be quickly completed by calling existing tools (e.g., spaCy), and is largely only applicable to the short queries analyzed in the paper.

2. In Section 6, the author proposes an application. However, the section is very unrigorous.
    - The claimed effectiveness of application lack systematic experiments (only two case studies are given). There is a complete absence of systematic, quantitative experiments on any standard benchmark. No pre- and post-optimization accuracy comparisons on a validation set are reported.
    - The section omit the details necessary for reproduce or validate the method. For instance: (a) How are the "high-dimensional embeddings of POS tags and dependency relations" computed? (b) What LLM was used to perform the task?
    - Also, the syntactic-based method is only sound for the short queries discussed in the paper, while is hard to scale to real complex prompts. For example, what is the "average syntax" of a system prompt that defines complex logic, tools, and constraints? How can the similarity to the centroid be calculated?
    - The method, as described by the author, use the collected datasets for defining the centroid, even without any selecting or filtering. This only results in a average quality centroid, not a high-quality centroid, which makes the section less reliable.

**Questions:**

See Weakness.

---

### Official Review · Reviewer_5erw · 2025-11-04

**Soundness:** 3
**Presentation:** 2
**Contribution:** 2
**Rating:** 2
**Confidence:** 4

**Summary:**

This paper, “Large Language Model Prompt Datasets: An In-Depth Analysis and Insights,” presents what appears to be the first systematic effort to collect, classify, and analyze datasets composed of prompts designed for large language models (LLMs). The authors gather 129 distinct prompt datasets totaling over 673 million prompts (≈1.22 TB), obtained from diverse sources such as Hugging Face, GitHub, Kaggle, and community repositories. They develop a hierarchical taxonomy for categorizing prompt datasets across multiple dimensions, including source, content, prompt-engineering methods, target application, and modality. The study then selects seven representative datasets (including Self-Instruct, ShareGPT, OASST1, dolly-15k, and medical-o1-reasoning-SFT) to perform multi-level analyses—lexical (n-grams), syntactic (dependency and POS structure), and semantic (Sentence-BERT embeddings)—comparing these to standard corpora such as EWT and ParTUT. The authors finally propose a prompt optimization technique that leverages syntactic centroid embeddings of POS and dependency features to improve prompt quality and LLM response accuracy.

**Strengths:**

The paper’s main contribution lies in assembling and systematizing an impressive collection of prompt datasets. The taxonomy (Figure 1) is detailed and potentially very useful as a reference for future work on prompt engineering and dataset analysis. The multi-level linguistic analysis (Sections 5.2–5.4) is comprehensive, combining traditional NLP tools (n-grams, POS, dependency, TF-IDF) with modern embedding-based semantics (Sentence-BERT). The inclusion of clear examples—e.g., comparison of high-frequency n-grams, dependency distributions, and PCA visualization of semantic embeddings—makes the findings interpretable. The proposed “prompt centroid optimization” is an original idea that links linguistic analysis to a practical downstream application, demonstrating at least one successful case (Figure 5). The dataset curation process is explicitly defined and adheres to clear inclusion criteria (dataset size, quality, relevance, accessibility). Overall, the paper offers a valuable empirical resource and a foundation for future research into the linguistic structures of prompts.

**Weaknesses:**

Despite the scale and ambition, several issues limit the technical depth and reproducibility. First, while the taxonomy is detailed, it is largely descriptive; there is no quantitative validation or inter-annotator reliability to demonstrate its robustness. Second, the “prompt optimization” approach is insufficiently evaluated—it is shown on a single illustrative case rather than through systematic benchmarking or quantitative improvements across tasks. The notion of aligning to a syntactic “centroid” is intriguing but lacks theoretical grounding or clear metrics of success (e.g., human evaluation, BLEU-based improvements, or task accuracy). Third, the linguistic analyses, though thorough, often remain surface-level: the paper shows frequency and ratio tables but stops short of statistical tests or model-based interpretations that would solidify claims about dataset similarity or diversity. In Section 5, for instance, the cosine similarity heatmap is interpreted qualitatively without quantifying variance or conducting clustering significance tests. Additionally, the 1.22 TB corpus is not clearly documented—only 129 datasets are listed, but the main text does not provide URLs, schema details, or licensing summaries beyond Appendix E. Given that accessibility and reproducibility are key to dataset-centric research, this omission weakens the contribution. Finally, while the authors claim the optimization method “improves meaningfulness,” no objective evaluation, ablation, or human-LLM study supports this.

**Questions:**

Clarify how the “centroid” embeddings are computed (simple average or weighted by TF-IDF, dataset size, etc.).

---

### Note · Authors · 2025-12-09

**Comment:**

Dear reviewers,

We appreciate your time and constructive feedback on our submission. We acknowledge that this paper requires heavy revision, thus we have decided to withdraw the manuscript to address the reviewers’ comments and prepare a revised version for a potential resubmission.

Best regards,
Authors

**Withdrawal Confirmation:**

I have read and agree with the venue's withdrawal policy on behalf of myself and my co-authors.